# From Complex to Atomic: Enhancing Augmented Generation via Knowledge-Aware Dual Rewriting and Reasoning

## Abstract

Recent advancements in Retrieval-Augmented Generation (RAG) systems have significantly enhanced the capabilities of large language models (LLMs) by incorporating external knowledge retrieval. However, the sole reliance on retrieval is often inadequate for mining deep, domain-specific knowledge and for performing logical reasoning from specialized datasets. To tackle these challenges, we present an approach, which is designed to extract, comprehend, and utilize domain knowledge while constructing a coherent rationale. At the heart of our approach lie four pivotal components: a knowledge atomizer that extracts atomic questions from raw data, a query proposer that generates subsequent questions to facilitate the original inquiry, an atomic retriever that locates knowledge based on atomic knowledge alignments, and an atomic selector that determines which follow-up questions to pose guided by the retrieved information. Through this approach, we implement a knowledge-aware task decomposition strategy that adeptly extracts multifaceted knowledge from segmented data and iteratively builds the rationale in alignment with the initial query and the acquired knowledge. We conduct comprehensive experiments to demonstrate the efficacy of our approach across various benchmarks, particularly those requiring multihop reasoning steps. The results indicate a significant enhancement in performance, up to 12.6% over the second-best method, underscoring the potential of the approach in complex, knowledge-intensive applications.

## 1 Introduction

Large Language Models (LLMs) have revolutionized the field of natural language processing by demonstrating the capability to generate coherent and contextually relevant text. These advanced models are trained on expansive corpora, equipping them with the versatility to execute a diverse spectrum of linguistic tasks, ranging from text completion to translation and summarization (Achiam et al., 2023; Bahrini et al., 2023; Touvron et al., 2023; Anil et al., 2023). Despite their broad capabilities, LLMs exhibit pronounced limitations when tasked with specialized queries in professional domains (Ling et al., 2024; Wang et al., 2023a), a demand that is particularly acute in practical applications. This primarily stem from the scarcity of domain-specific training material and an limited grasp of private knowledge and rationale within these domains. As a result, LLMs may produce responses that are not only potentially erroneous but also lack the detail and precision required for expert-level engagement (Bender et al., 2021). Besides the limitations in the domain-specific tasks, another striking issue with LLMs is the phenomena known as "hallucination", where the model generates information that is not grounded in reality or factual data (Beltagy et al., 2020; Xu et al., 2024). Moreover, the knowledge base of LLMs, being static and crystallized at the point of their last update, introduces temporal stasis (Brown et al., 2020). Further compounding these challenges is the issue of long-context comprehension (Li et al., 2024). Existing LLMs struggle to maintain an understanding of task definitions across long context, and their performance tends to deteriorate significantly when confronted with more complex and demanding tasks.

To mitigate these issues, Retrieval-Augmented Generation (RAG) (Lewis et al., 2020) has emerged as a promising solution, augmenting LLMs with external knowledge retrieval to anchor generated content in factual data. While RAG systems improve the accuracy and reliability of generated re-

sponses by incorporating relevant external information, they often fall short in handling tasks that demand deep domain-specific reasoning. Current RAG methods predominantly rely on text retrieval, without explicitly focusing on extracting and understanding the underlying knowledge needed to tackle complex, multihop reasoning tasks. In this work, we argue that the key to advancing RAG lies in knowledge-aware processing, where domain-specific knowledge is not only retrieved but also atomized, contextually-decomposed, and reasoned over in a more structured and adaptive manner.

**The Importance of Knowledge-aware Processing** Addressing complex, logic-driven tasks in specialized domains requires more than surface-level retrieval of relevant text passages. It demands knowledge extraction and comprehension to deeply understand both the user's information needs and the underlying context of the retrieved data. For example, specialized queries in fields like medicine, law, or finance often involve domain-specific terminology and logic, which generic LLMs may fail to grasp fully. Traditional RAG systems that retrieve text passages based on keyword matching (Ram et al., 2023; Jiang et al., 2023) or embedding similarity (Gao et al., 2023) may retrieve contextually relevant but semantically shallow information, insufficient for answering intricate questions. The challenge lies in ensuring that the retrieved knowledge aligns with the true intent behind the user's query and the broader context of the problem at hand.

Moreover, effective comprehension of knowledge extends beyond extracting factual data. It also involves understanding the semantics and relationships hidden within the data corpus. To this end, some recent efforts (Raina & Gales, 2024) attempted to rephrase the original questions with integrating hypothestical answers (Gao et al., 2022) or transform text passages into lists of simple yet atomic questions (Raina & Gales, 2024). This is especially crucial when multiple sources of information must be integrated to form a coherent response. Without a mechanism to extract and comprehend knowledge from each chunk of domain data, current RAG systems are limited in their ability to handle complex tasks that require deep contextual understanding.

**The Need for Iterative Reasoning in multihop Questions** In many real-world scenarios, a single retrieval or answer generation step may not be enough to fully resolve a complex query. multihop reasoning tasks, where the answer depends on synthesizing information from multiple sources, demand the decomposition of the original query into a series of simpler, interrelated subquestions (Press et al., 2023). Nonetheless, this approach may face obstacles in domains where the knowledge is not readily accessible to LLMs. We argue that the decomposition in such domains should be contextual, rather than a standalone operation, meaning that decomposed queries can be answered with the retrieved knowledge and context progressively and evolve into refining subsequent queries. This iterative approach allows the system to evolve its understanding of the user's inquiry, ensuring that follow-up questions are informed by the most recent retrieval results.

For example, consider the HotpotQA (Yang et al., 2018) dataset question: "Who was born first, Erika Jayne or Marco Da Silva?". A straightforward rewriting strategy might decompose this into two separate queries: "What is the age of Erika Jayne?" and "What is the age of Marco Da Silva?". However, if the available knowledge base only contains information such as "Her father left in 1971 when Erika Jayne was 9 months old.", traditional rewriting may not lead to the retrieval of relevant information. Techniques like Think-on-Graph (ToG) (Sun et al., 2024), which apply beam search and iterative reasoning on knowledge graphs (KGs), could help. Yet, the performance of KG-based methods is often hampered by limited richness of the knowledge within their structured triples and the need for a high-quality, domain-specific KG. For instance, GPT-4 might generate triples like (Her father, left, action) and (Erika Jayne, 9 months old, age), which are insufficient for reconstructing the original context, "Her father left in 1971 when Erika Jayne was 9 months old." [1].

To overcome these limitations, we introduce a novel framework, KAR³-RAG, that employs a knowledge-aware dual rewriting and reasoning mechanism. Our approach features a dynamic interaction between question rewriting and knowledge retrieval, enabling the system to adaptively refine both the query and the retrieved context at each iteration. The core components of our system include Knowledge atomizer, decomposing raw data into atomic questions for more granular retrieval, Query proposer, generating follow-up questions based on the evolving context, Atomic

---

[1] We use GPT-4-1106-preview and the KG construction prompt from `https://github.com/rahulnyk/knowledge_graph` to extract entity relations from the statement "Her father left in 1971 when Erika Jayne was 9 months old,", it generates six triples as follows: "(Her father, left, action), (Her father, 1971, time), (Her father, Erika Jayne, relation), (Erika Jayne, 9 months old, age), (Erika Jayne, 1971, time), (1971, 9 months old, time relation)"

retriever, identifying and retrieving relevant knowledge based on atomic knowledge alignments, and Atomic selector, determining the most relevant follow-up questions based on the retrieved information. By leveraging these components, our system can iteratively refine its understanding of both the question and the retrieved knowledge, enabling more accurate and context-aware reasoning over multiple hops.

Our key contributions are as follows: 1). We present a novel RAG framework that capitalizes on the synergistic interaction between interdependent rewriting and reasoning processes, ensuring full utilization of the available context. 2). We enhance retrieval efficacy through a dual rewriting mechanism that modifies both the original questions and the text passages (chunks). Our reasoning process is context-aware, enabling the adaptive formulation of follow-up questions based on the provided context. 3). We report on comprehensive experimental and ablation studies that validate the superior performance of our approach across multiple benchmark datasets, which is up to 12.6% increase over the second-best method.

## 2 RELATED WORK

### 2.1 RAG

Retrieval-Augmented Generation (RAG) has emerged as a promising solution that effectively incorporates external knowledge to enhance response generation. With the booming of LLMs (Bahrini et al., 2023; Touvron et al., 2023), most research in the RAG paradigm has shifted towards a framework that initially retrieves pertinent information from external data sources and subsequently integrates it into the context of the query prompt as supplementing knowledge for contextually relevant generation (Ram et al., 2023). To enhance the retrieval quality of the naive RAG, advanced RAG approaches implement specific enhancements across the pre-retrieval, retrieval, and post-retrieval processes, including query optimization (Ma et al., 2023; Zheng et al., 2023), multi-granularity chunking (Chen et al., 2023; Zhong et al., 2024), mixed retrieval (Yang, 2023) and re-ranking (Cohere, 2023). On one hand, efforts focus on query rewriting, either explicitly (Zheng et al., 2024) or implicitly (Gao et al., 2022), to enhance retrieval performance. On the other hand, several studies transform raw data sources into structured data, ultimately converting them into valuable knowledge for more effective retrieval and reasoning(Wang et al., 2023b; Zheng et al., 2024; Raina & Gales, 2024). In our system, we introduce atomic rewriting for both queries and chunks, which not only achieves multi-granularity query decomposition but also comprehensively extract inherent knowledge from chunks. It has been recognized that naive RAG systems is insufficient for tackling complex tasks such as summarization (Hayashi et al., 2021) and multihop reasoning (Ho et al., 2020). Consequently, most recent research focuses on developing advanced coordination schemes that leverage existing RAG modules to collaboratively address these challenges. Iter-RetGen (Shao et al., 2023) and DSP (Khattab et al., 2023) employ retrieve-read iteration to leverage generation response as the context for next round retrieval. FLARE (Jiang et al., 2023) propose a confidence-based active retrieval mechanism that dynamically adjusts query during iterative retrieval processes. Our approach adopts an iteration-based RAG pipeline that leverages context-aware reasoning process, enabling the adaptive formulation of follow-up questions for each iteration and reducing the difficulty of retrieval and reasoning of complex tasks.

### 2.2 MULTIHOP QA

multihop Question Answering (MHQA) (Yang et al., 2018) involves answering questions that require reasoning over multiple pieces of information, often scattered across different documents or paragraphs. This task presents unique challenges as it necessitates not only retrieving relevant information but also effectively combining and reasoning over the retrieved pieces to arrive at a correct answer. The traditional graph-based methods in MHQA solves the problem by building graphs and inferring on graph neural networks(GNN) to predict answers (Qiu & other authors, 2019; Fang & other authors, 2020). With the advent of LLMs, recent graph-based methods (Li & Du, 2023; Panda et al., 2024) have evolved to construct knowledge graphs for retrieval and generate response through LLMs. Another branch of methods dynamically convert multihop questions into a series of subqueries by generating subsequent questions based on the answers to previous ones (Trivedi et al., 2023; Khattab et al., 2023; Feng et al., 2023). The subqueries guides the sequential retrieval and the retrieved results in turn are used to improve reasoning. Treating MHQA as a supervised problem,

Figure 1: Overview of the KAR$^3$-RAG workflow, illustrating knowledge atomizing by the atomizer, and knowledge-aware task decomposition using the query proposer, atomic retrieval and atomic selector. The query proposer generates atomic query proposals based on the original query and reference context. These proposals are used to retrieve the relevant atomic questions, producing retrieved atomic pairs. The atomic selector chooses the most relevant pair and the corresponding chunk, which is added to the reference context for task decomposition in the subsequent iteration. Once the atomic selector determines that no further information is required and no atomic pair are selected, the query and reference context are passed to the generator to produce the final answer.

Self-RAG (Zhang et al., 2024) trains an LM to learn to retrieve, generate, and critique text passages, and beam-retrieval (Asai et al., 2023) models the multihop retrieval process in an end-to-end manner by jointly optimizing an encoder and classification heads across all hops. Self-Ask (Press et al., 2023) improves CoT by explicitly asking itself follow-up questions before answering the initial question. This method enables the automatic decomposition of questions and can be seamlessly integrated with retrieval mechanisms to tackle multihop QA.

## 3 METHODOLOGY

### 3.1 PRELIMINARY

In a RAG system, the textual corpus is divided into a collection of document chunks, denoted as $\mathcal{D} = \{d_1, d_2, \ldots, d_n\}$, where $d_i$ represents the $i$-th document chunk. The original query is denoted as $q$, and its corresponding ground truth answer is represented by $a$. The retrieval phase involves evaluating the similarity between the query $q$ and each document chunk $d_i$, after which the top-$k$ most relevant chunks are selected as retrieval results, forming the basis for subsequent generation.

$$\mathcal{R} : \operatorname*{topk}_{d_i \in \mathcal{D}} \operatorname{Sim}(q, d_i) \to D^q \tag{1}$$

Here, the retriever $\mathcal{R}$ selects the top-$k$ most relevant chunks $D^q$ based on the similarity function $Sim(\cdot)$. Finally, the original query and retrieved chunks are fed into the large language model to generate the answer, denoted as $\hat{a} = \mathcal{LLM}(q, D^q)$. In the advanced RAG systems, query rewriting is employed to bridge the semantic gap between the query and the chunks to be retrieved. The rewritten query is represented as $\hat{q} = f_{re}(q)$. The workflow of the advanced RAG system is further improved as follows,

$$\hat{a} = \mathcal{LLM}(q, D^{\hat{q}}), \text{ where } D^{\hat{q}} = \mathcal{R}(\hat{q}, \mathcal{D}) \tag{2}$$

This enhancement allows the system to better align queries with relevant document chunks, enhancing retrieval accuracy and answer generation. However, addressing complex multihop questions remains challenging. These questions often require reasoning across multiple chunks and integrating information through several retrieval and generation steps-a process that a single pass may not fully capture.

### 3.2 FRAMEWORK

To address complex multihop questions, we introduce an enhanced RAG system with **K**nowledge-**A**ware **dual R**ewriting and **R**easoning, termed as KAR$^3$. This system employs an iterative retrieval-reasoning-generation mechanism that facilitates gradual collection of relevant information and progressive reasoning over incremental context. An overview of the proposed workflow is depicted

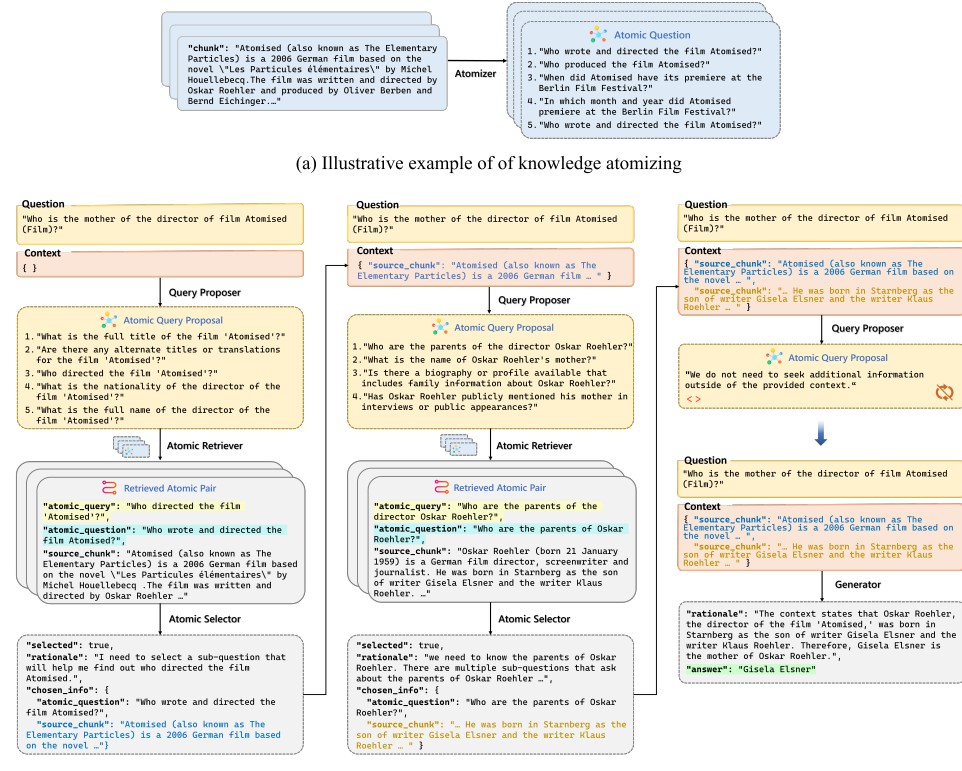

(a) Illustrative example of of knowledge atomizing

(b) Illustrative example of KAR$^3$-RAG case

Figure 2: Illustrative examples of KAR$^3$-RAG cases: (a) Example of knowledge atomizing, (b) RAG case with knowledge-aware task decomposition. As iterations progress, the reference context is enriched by accumulating relevant chunks through atomic retrieval and selection. With the expansion of context, the number of atomic query proposals generated decreases until no further proposals are produced. Subsequently, the iteration process terminates, and the combined query and context are harnessed to produce the final response.

in Figure 1. In our framework, raw data chunks are broken down into atomic questions using a knowledge atomizer to construct atomic knowledge base for the subsequent retrieval. Queries are similarly atomized by a query proposer to generate atomic query proposals, which are utilized to retrieve the relevant atomic questions from the knowledge base. Both chunks and queries are rewritten to bridge the semantic gap and improve the alignment of knowledge. An atomic retriever then selects the top-$k$ atomic pairs for each atomic query proposal. Based on these retrieved atomic pairs, an atomic selector, as a reasoner, identifies the most useful atomic pair for problem-solving and adds the corresponding raw chunk to the context. This context is then aggregated with query for the task decomposition in next iteration. The iteration process may terminate earlier if it fails to retrieve suitable atomic questions, either due to the generation of low-quality question proposals or the lack of relevant atomic question candidates. At this point, the query and context are passed to the generator to produce the final answer.

### 3.3 KNOWLEDGE ATOMIZING

Chunked text often contains multifaceted information, and typically, only a subset is needed to address a specific task. Traditional information retrieval methods, which consolidate all information within a single chunk may not facilitate the efficient retrieval of the precise information required. Recent research have explored the extraction of triple knowledge units from chunked text and constructing knowledge graphs to facilitate efficient information retrieval (Edge et al., 2024; Panda et al., 2024). However, the construction of these knowledge graphs is costly, and the inherent knowledge may not always be fully explored. To better present the knowledge embedded in documents, we propose atomizing the original documents for knowledge extraction, a process we refer as *Knowledge*

*Atomizing*. This approach leverage the context understanding and content generation capabilities of LLMs to automatically tag atomic knowledge pieces within each document chunk.

The presentation of the atomic knowledge can be various. Instead of utilizing declarative sentences or subject-relationship-object tuples, we propose using questions as knowledge indexes to further bridge the gap between stored knowledge and query. In knowledge atomizing process, we input the document chunk to an LLM as context, ask it to generate relevant questions that can be answered by the given chunk as many as possible. These generated atomic questions are stored together with the given chunks. The knowledge atomizer applies atomizing operation on each chunk.

$$f_a(d_k) = \{q_{k1}, q_{k2}, \cdots, q_{km}\} \tag{3}$$

The atomic questions are generated by atomizer for every chunk, forming an atomic knowledge base, denoted as $\mathcal{KB} = \{f_a(d_k), d_k\}$. An example of knowledge atomizing is illustrated in Figure 2(a), where the atomic questions encapsulate various aspects of the knowledge contained within the chunk. Since each chunk is tagged with multiple atomic questions, an atomic query can be used to locate relevant atomic questions, which then leads to the associated reference chunks.

## 3.4 KNOWLEDGE-AWARE TASK DECOMPOSITION

Addressing complex multihop questions often requires integrating multiple pieces of knowledge, which implicitly demands the ability to break down the original question into several sequential or parallel atomic questions for retrieval. We refer to this operation as *Task Decomposition*. By combining the extracted atomic knowledge with the original chunks, we construct an atomic knowledge base. Each time a task is decomposed, the atomic knowledge base provides insights into the available knowledge, enabling knowledge-aware task decomposition. We design the *Knowledge-Aware Task Decomposition* workflow, and the complete algorithm for solving task is detailed in Algorithm 1, and an example is illustrated in Figure 2(b).

Initially, the reference context $\mathcal{C}_0$ is initialized as an empty set. In the first iteration, task decomposition relies solely on the query to generate atomic query proposals. As iterations progress, the accumulated context at $t$-th iteration denoted as $\mathcal{C}_{t-1}$, consists of chunks retrieved from previous iterations. During the $t$-th iteration, the query proposer generates atomic query proposals based on the original query and the accumulated context.

$$f_p(q, \mathcal{C}_{t-1}) = \{\hat{q}_1^t, \hat{q}_2^t, \cdots, \hat{q}_n^t\} \tag{4}$$

The query proposer $f_p(\cdot)$ can be implemented as either an LLM or a learnable component. we leverage an LLM to generate query proposals that are potentially beneficial for task completion, represented as $\hat{\boldsymbol{q}}^t = \{\hat{q}_i^t\}$. During this process, the selected reference chunks $\mathcal{C}_{t-1}$ are provided as context to avoid generating proposals linked to already known knowledge. Consequently, the query proposals evolve with each iteration, adapting to the updated context and aiming to explore additional knowledge beyond chunks in the context. For each atomic question proposal, we retrieve its top-$k$ relevant atomic question candidates along with their source chunks from the knowledge base. The atomic retrieval process is:

$$\mathcal{R}_{atom} : \mathop{\mathrm{topk}}_{q_{kl} \in f_a(\mathcal{D})} \mathrm{Sim}(\hat{q}_i^t, q_{kl}) \xrightarrow{\mathcal{KB}} P^{\hat{q}_i^t} \tag{5}$$

where the atomic retriever, denoted as $\mathcal{R}_{atom}$, produces a set of retrieved atomic pairs for each atomic query proposal, represented as $P^{\hat{q}_i^t} = \{(\hat{q}_i^t, q_{k_i l_i}, d_{k_i})\}$. All the retrieved atomic pairs from each atomic query proposal are aggregated to generate an overall set $P^{\hat{\boldsymbol{q}}^t}$. We employ cosine similarity of the corresponding embeddings to retrieve the top-$k$ atomic questions, provided their similarity to a proposed atomic question meets or exceeds a specified threshold $\delta$. With the original question, the accumulated context, and the list of retrieved atomic pairs, the atomic selector employ an LLM to select the most useful atomic pair for problem-solving.

$$\mathcal{LLM}(q, \mathcal{C}_{t-1}, P^{\hat{\boldsymbol{q}}^t}) = (\hat{q}_s^t, q_{k_s l_s}, d_{k_s}) \tag{6}$$

The atomic selector, denoted as $\mathcal{S}_{atom}$, further retrieve the relevant raw chunk of the atomic pair selected as the new context added in the $t$-th iteration, denoted as $c_t$. This chunk corresponds to $d_{k_s}$

---

**Algorithm 1** Task Solving with Knowledge-Aware Decomposition

---

1: Initialize context $\mathcal{C}_0 \leftarrow \phi$
2: **for** $t = 1, 2, \ldots, N$ **do**
3:     Generate atomic question proposals $\hat{q}^t \leftarrow f_p(q, \mathcal{C}_{t-1})$
4:     Retrieve top-$k$ atomic pairs for each atomic query proposal from knowledge base

$$P^{\hat{q}^t} \xleftarrow{\mathcal{KB}} \mathcal{R}_{atom}(\hat{q}^t, f_a(\mathcal{D}))$$

5:     Select the most useful atomic question or *None* when additional information is unnecessary

$$q_{k_s l_s} \leftarrow \mathcal{LLM}(q, \mathcal{C}_{t-1}, P^{\hat{q}^t})$$

6:     **if** $q_{k_s l_s}$ is *None* **then**
7:         $\mathcal{C}_t \leftarrow \mathcal{C}_{t-1}$
8:         **break**
9:     **else**
10:         Fetch the relevant chunk $c^t$ corresponding to $q_{k_s l_s}$
11:         Update context $\mathcal{C}_t \leftarrow \mathcal{C}_{t-1} \cup c^t$
12:     **end if**
13: **end for**
14: Generate answer $\hat{a} \leftarrow \mathcal{LLM}(q, \mathcal{C}_t)$

---

in equation 6. The chunk retrieval process can be represented by the following formula,

$$c_t = \mathcal{S}_{atom}(\mathcal{R}_{atom}(f_p(q, \mathcal{C}_{t-1}), f_a(\mathcal{D}))) \tag{7}$$

This retrieved chunk is aggregated into the reference context for the next round of decomposition, expressed as $\mathcal{C}_t = c_t \cup \mathcal{C}_{t-1}$. Knowledge-aware decomposition can iterate up to $N$ times, where $N$ is a hyperparameter set to control computational cost. The iteration process may conclude earlier if it fails to retrieve suitable atomic questions, either due to the generation of low-quality question proposals or the absence of relevant atomic question candidates. Alternatively, the process can be halted if the $\mathcal{LLM}$ deems the accumulated knowledge adequate for task completion. This early termination mechanism allows the process to conclude before completing all iterations, reducing computational costs without compromising accuracy. Finally, the accumulated context $\mathcal{C}_t$ is utilized to generate answer $\hat{a}$ for the given query $q$ in line 14.

It is worth mentioning that the knowledge-aware decomposition can be a learnable component. For each private knowledge base, we can utilize the data collected in each decomposition iteration—specifically $(q, a, \hat{a}, \{\hat{q}_s^t, c^t, \hat{q}^t, P^{\hat{q}^t}, \mathcal{C}_t\})$. This trained proposer can then directly suggest atomic queries $q^t$ during inference, which means lines 3 to 5 in Algorithm 1 can be replaced by a single call to this learned proposer, thereby reducing both inference time and computational cost. We leave the exploration of training an efficient query proposer as future work.

## 4 EVALUATION AND METRICS

The experimental setup is detailed in Section 4.1, while the primary experimental results are outlined in Section 4.2. Ablation studies are discussed in Section 4.3. Additionally, evaluations on two legal domain-specific benchmarks and three real case studies are included in Appendix A.4 and Appendix A.5, respectively, due to content constraints.

### 4.1 EXPERIMENTAL SETUP

**Methods** To thoroughly evaluate the performance of our proposed knowledge-aware decomposition approach, we have selected a variety of baseline methods that represent different strategies for task-solving with LLMs. We include **Zero-Shot CoT**(Kojima et al., 2022) to assess the inherent reasoning capabilities and built-in knowledge of the underlying LLM without any additional context. **Naive RAG**(Lewis et al., 2020), which introduces external knowledge through retrieval, serves as a benchmark for evaluating the incremental benefits of augmented knowledge. The **Self-Ask** framework(Press et al., 2023) is employed to investigate the impact of an iterative question decomposition

and answering strategy on task performance. The **IRCoT**(Trivedi et al., 2023), which iteratively generates the rationale to process the multihop questions, and the **Iter-RetGen**(Shao et al., 2023), which iteratively uses the recent response as a retrieval query to improve the response quality, are also conducted for performance comparison. Detailed descriptions of these experimental methods are provided in Appendix A.3, and below are the brief summaries: **Zero-Shot CoT**: Questions are addressed using solely the Chain-Of-Thought (CoT) technique without any example demonstrations or supplemental context.**Naive RAG**: This approach employs dense retrieval from a flat knowledge base to procure relevant information for each question as the context in question answering. **Self-Ask w/ Retrieval**: This method employs a task decomposition strategy wherein the LLMs is prompted to iteratively generate and answer follow-up questions. Furthermore, naive RAG is applied for answering each follow-up question. **IRCoT**: This approach iteratively prompts LLMs to generate one more sentence of rationale with retrieved passages, and retrieves new passages with the newly generated reason. **Iter-RetGen**: This method iteratively answers questions with retrieved passages, and uses the newly generated rationale and answer for the next-round retrieval. **KAR³**: The proposed approach that iteratively decomposes complex questions into sub-questions and retrieves relevant knowledge.

In our experiments, we employ GPT-4 (1106-Preview version) across all the methods outlined previously. For the experiments presented in Section 4.2, the iteration number $N$ is set to 5 for Self-Ask with Retrieval, IRCoT, Iter-RetGen and KAR³. Additionally, the atomic retriever is initialized with $k = 4$ and $\delta = 0.5$. A comprehensive list of hyper-parameters for the retrieval and LLM can be found in Appendix A.2.

**Metrics**   To ensure consistency with established benchmarks, we adopt **F1** as a conventional metric in our experimental evaluation. To more accurately assess the the alignment of responses with the intended answers—beyond mere lexical matching—we introduce a novel evaluation metric employing *GPT-4*. In this process, *GPT-4* acts as an evaluator, assessing the correctness of a response in relation to the question and the correct answer labels. We refer to this metric as **Accuracy (Acc)**. Upon manual inspection of a sample set, the judgments rendered by *GPT-4* demonstrate complete agreement with human evaluators, affirming the reliability of this metric.

Specifically, in cases where multiple correct answer labels are available, we employ a conservative scoring approach for **F1** by retaining the highest score achieved. While in the context of computing **Accuracy (Acc)**, all admissible answer labels are furnished concurrently to the evaluation process, resulting in a singular accuracy score. Furthermore, a full evaluation results with Exact Match (EM), Recall and Precision can be found in Appendix A.3.

**Datasets**   Since we are targeting at solving multihop reasoning tasks, three widely-recognized multihop datasets: HotpotQA(Yang et al., 2018), 2WikiMultiHopQA(Ho et al., 2020), and MuSiQue(Trivedi et al., 2022) are used in our evaluation. A brief introduction to these datasets can be found in Appendix A.1. For each dataset, we randomly sample 500 QA data from the *dev* set, disregarding the question type and the number of hops to ensure randomness. We compile the context paragraphs from all sampled QA data into a single knowledge base for each benchmark, creating a more complex retrieval scenario. This design choice aims to rigorously assess the task decomposition and relevant context retrieval capabilities of our model.

## 4.2 MAIN RESULTS

As demonstrated in Table 1, our approach achieves superior performance across all datasets, yielding approximately $+1.4(1.6\%)$, $+7.2(9.6\%)$ and $+7.0(12.6\%)$ increases in accuracy over the second best results for HotpotQA, 2WikiMultiHopQA and MuSiQue, respectively. For brevity, 2WikiMultiHopQA is abbreviated as 2Wiki in the result tables.

When comparing the performance of Zero-Shot CoT and Naive RAG, the inclusion of retrieved context significantly boosts accuracy, with gains ranging from $+10.03(42.7\%)$ to $+29.0(54.1\%)$. Furthermore, by incorporating decomposition mechanisms-either by asking follow-up questions as in Self-Ask w/ Retrieval method or by reasoning step-by-step as in IRCoT approach-performance can be further improved. Although both IRCoT and Iter-RetGen utilize rationale as the retrieval query to minimize the semantic gap, Iter-RetGen outperforms IRCoT, as shown in Table 1. A key distinction between these two methods is that IRCoT uses only the newly generated rationale sen-

Table 1: Performance comparison on multihop QA datasets. Best in bold, second-best underlined.

| Method | HotpotQA | | 2Wiki | | MuSiQue | |
|---|---|---|---|---|---|---|
| | F1 | Acc | F1 | Acc | F1 | Acc |
| Zero-Shot CoT | 43.94 | 53.60 | 41.40 | 43.87 | 22.90 | 23.47 |
| Naive RAG | 72.67 | 82.60 | 59.74 | 62.80 | 43.31 | 44.40 |
| Self-Ask w/ Retrieval | 71.40 | 80.00 | 69.06 | 75.00 | 46.76 | 51.40 |
| IRCoT | 67.30 | 81.00 | 63.83 | 70.40 | 47.57 | 49.20 |
| Iter-RetGen | 75.27 | 86.60 | 67.21 | 73.60 | 52.48 | 55.60 |
| KAR³ (Ours) | **76.48** | **88.00** | **75.00** | **82.20** | **57.86** | **62.60** |

tence as the query, without revisiting previously generated rationales, while Iter-RetGen uses the entire rationale generated in last round as the retrieval query, allowing for the reevaluation or correction of past rationales. This suggest that incorporating a mechanism for rethinking or correcting historical generations may be critical for enhancing performance.

Our proposed approach, KAR³, emphasizes knowledge-aware task decomposition and differs from the spontaneous decomposition mechanism reliant on given demonstrations, as employed by Self-Ask. It performs decomposition with an awareness of available knowledge and effectively uses atomic questions as an intermediate medium to bridge the semantic gap. The "proposal first, then select" framework, detailed in Algorithm 1, provides an opportunity to validate the intent of the question and rectify potential errors in the historical rationale generation process. A practical application of this point can be seen in Case(a) of Appendix A.5. Consequently, the experimental results demonstrate that KAR³ consistently outperforms other methods, validating its effectiveness in complex reasoning scenarios.

### 4.3 ABLATION STUDY

**The selection of $N$.** We first conducted experiments with the iteration upper bound $N$ set to $1, 2, \ldots 10$, and the results are presented in Figure 3. Detailed performance metrics are available in Table 6 of Appendix A.3. Across all three datasets, there is a consistent uptrend in both Supporting Fact Recall and Answer Accuracy. This pattern underscores the approach's capability to incrementally enhance its outputs through additional iterations, particularly when more detailed and contextually relevant information is required to address problem.

Additionally, upon examining the relationship between the number of iterations and the observed growth in supporting fact recall, we note that for HotPotQA and 2WikiMultiHopQA datasets, the recall curves exhibit a pronounced increase up to the fourth iteration. Conversely, the recall for the MuSiQue dataset continues to rise sharply beyond this point, even though the maximum number of hops per question is capped at four, as mentioned in Appendix A.1. This discrepancy implies that while KAR³ is adept at retrieving relevant and useful information within a limited number of iterations, it still has certain limitation: KAR³ relis on the reasoning capability of the used LLM, and further iterations may be required to fully capture the necessary information, especially as the complexity of the questions increases.

Although the algorithm, as outlined in Algorithm 1, does incorporate early-stopping mechanisms that prevent every question from reaching the maximum iteration limit, a higher $N$ invariably leads to increased computational demands. Therefore, the selection of an appropriate $N$ calls for a delicate balance between computational resources and the expected enhancement in performance. To this end, we choose $N = 5$ for the experiments in Section 4.2. This value is slightly above the maximum number of hops in the datasets and is justified by the plateauing of performance gains beyond this point as evidenced in Figure 3. It reflects a pragmatic trade-off that accounts for both the computational cost and the retrieval efficacy of our approach.

**The variants of approach components.** KAR³ is comprised of four key components: (a) a knowledge atomizer, (b) a query proposer, (c) an atomic retriever, and (d) an atomic selector. We conduct ablation studies to ascertain the individual and collective contributions of these components by introducing several method variants. For the query proposer, the **Single Proposer** variant assesses

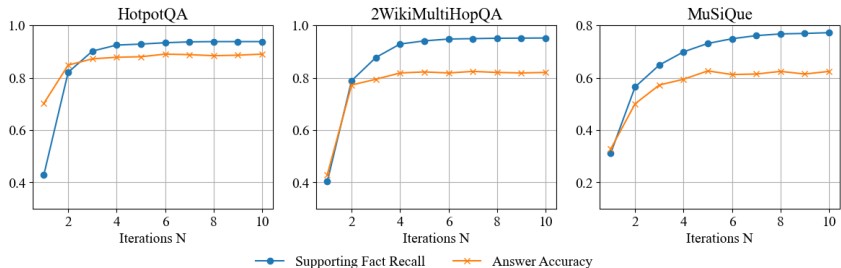

Figure 3: Supporting fact recall and answer accuracy over iterations. Supporting fact recall is depicted in blue, while answer accuracy in orange.

Table 2: Ablation study on components of KAR$^3$.

| Variant Method | HotpotQA | | 2Wiki | | MuSiQue | |
|---|---|---|---|---|---|---|
| | F1 | Acc | F1 | Acc | F1 | Acc |
| w/ Single Proposer | 75.06 | 85.60 | 70.19 | 76.40 | 49.67 | 52.20 |
| w/ Chunk Selector | 72.80 | 83.20 | 61.65 | 65.80 | 49.31 | 53.40 |
| w/ Chunk Retriever | 76.31 | 86.60 | 67.14 | 72.40 | 49.05 | 53.00 |
| KAR$^3$ (Ours) | 76.48 | 88.00 | 75.00 | 82.20 | 57.86 | 62.60 |

the impact of generating only a single query as opposed to multiple queries. In the case of the atomic selector, the **Chunk Selector** variant examines the implications of selecting information in larger segments, or chunks, rather than focusing on atomic questions. Finally, the **Chunk Retriever** variant combines the modifications of the previous two variants: it generates a single query and retrieves information in the form of chunks from a knowledge base that has not been pre-processed into atomic units. The atomic selection phase, corresponding to Algorithm 1 line 5, is then replaced by directly selecting the most useful chunk since no available atomic question in this variant. These variants enable us to isolate the impact of each component and understand how they interact to produce the system's output. The experimental results of these variants are presented in Table 2.

As evidenced by the results in Table 2, the individual contributions of the components were evaluated. We observed that limiting the approach to propose only a single atomic query led to accuracy reductions of 2.8%, 7.0% and 16.6% for the respective datasets. Similarly, opting for chunk-based selections over atomic questions resulted in Accuracy declines of 5.5%, 16.2% and 14.7%. The substitution of the atomic retriever with a general chunk retriever caused the Accuracy to drop by approximately 1.6%, 11.9%, 15.3%, respectively. These ablation studies imply that each designed component is crucial for achieving optimal retrieval performance and coherent reasoning traces.

**Limitation Discussion.** Beyond the need for additional iterations to extract crucial information for complex questions, our experiments with GPT-3.5 (details in Table7 in AppendixA.3), indicate a limitation in relying on LLMs' reasoning capabilities. The performance of KAR$^3$ does not significantly surpass that of methods like IRCoT and Self-Ask w/ Retrieval and occasionally falls short compared to Self-Ask w/ Retrieval. This highlights that KAR$^3$'s success hinges on its advanced reasoning skills and its ability to robustly follow complex instructions.

## 5 CONCLUSION

We present an advanced RAG system, enhanced with knowledge-aware dual rewriting, and reasoning capabilities, designed to improve knowledge extraction and rationale formulation within specialized datasets. The comprehensive results from extensive experiments underscore the efficacy of our approach, particularly in scenarios involving benchmarks with multihop questions. For future work, we aim to refine the system's proficiency through the integration of in-context learning (Wei et al., 2022), by adaptively selecting demonstrations for the query proposer. This will further enhance its ability to perform knowledge-aware question rewriting. Additionally, we are interested in developing a knowledge-aware atomizer capable of incorporating feedback from sample questions, thereby improving its understanding of the most beneficial types of atomic knowledge.

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

# A  APPENDIX

## A.1  INTRODUCTION TO OPEN-DOMAIN BENCHMARKS

We provide a brief overview of the multihop QA datasets used in our experiments, noting that our method does not leverage the question type information nor the number of hops information during the solving process, as our approach is designed to be agnostic to such classifications. Table 3 outlines the distribution of question types within our sampled sets, offering insight into the variety of reasoning challenges presented in our evaluation, though this does not directly impact our method.

**HotpotQA**   The HotpotQA dataset is a well-known multihop QA benchmark primarily consisting of 2-hop questions, each associated with 10 Wikipedia paragraphs. Among these, some paragraphs contain supporting facts essential to answering the question, while the rest serve as distractors. The dataset also includes a *question type* field, which delineates the logical reasoning required—*comparison* questions involve contrasting two entities, and *bridge* questions require inferring the bridge entity, or inferring the property of an entity through an intermediary entity, or locating the answer entity (Yang et al., 2018). Although our method operates independently of these types, their description here is to exemplify the nature of questions within the dataset and to contextualize the expected performance variance across different benchmarks.

**2WikiMultiHopQA**   Inspired by HotpotQA, 2WikiMultiHopQA expands the diversity of question types. It retains the *comparison* type from HotpotQA and introduces *inference* and *compositional* questions that evolve from the *bridge* type by focusing on entity attribute deduction and entity location, respectively. Additionally, the *bridge comparison* type is a novel category that requires a synthesis of *bridge* and *comparison* reasoning. This dataset typically presents 2-hop to 4-hop questions, each accompanied by 10 Wikipedia paragraphs containing supporting facts and distractors. While these types inform the dataset's structure, they are not utilized by our method, which treats all questions uniformly regardless of their categorization. For the sake of brevity, 2WikiMultiHopQA is abbreviated to 2Wiki in the result tables throughout this paper.

**MuSiQue**   Addressing the issue that many multihop questions can be solved via short-cuts—arriving at correct answers without proper reasoning—MuSiQue implements stringent filters and additional mechanisms specifically designed to encourage connected reasoning, as reported by Trivedi et al. (Trivedi et al., 2022). Unlike the other datasets, MuSiQue does not categorize questions by type, but it does provide explicit information on the number of hops required for each question, ranging from 2 to 4 hops. Each question is associated with 20 context paragraphs, which introduce a mix of relevant and irrelevant information, further complicating the task of discerning the correct reasoning path. This explicit hop information, while not used by our method, underscores the complexity of the dataset and the robustness required by models to handle such challenges effectively.

Table 3: Distribution of question types across three distinct multihop QA datasets.

| Type | Count | Ratio |
|---|---|---|
| comparison | 132 | 26.4% |
| inference | 64 | 12.8% |
| compositional | 196 | 39.2% |
| bridge_comparison | 108 | 21.6% |

| Type | Count | Ratio |
|---|---|---|
| comparison | 107 | 21.4% |
| bridge | 393 | 78.6% |

| #Hops | Count | Ratio |
|---|---|---|
| 2 | 263 | 52.6% |
| 3 | 169 | 33.8% |
| 4 | 68 | 13.6% |

(a) HotPotQA     (b) 2WikiMultiHopQA     (c) MuSiQue

## A.2  HYPER-PARAMETERS

During the knowledge extraction phase, we utilize a *temperature* setting of 0.7 specifically for the *Knowledge Atomizing* process, promoting a balance between diversity and determinism in the generated atomic knowledge. Conversely, for all question-answering (QA) steps in each method, we implement a *temperature* of 0, ensuring consistent responses from the model.

Regarding the retrieval component, we engage the *text-embedding-ada-002* (version 2) as our embedding model for both the general knowledge bases and the atomic knowledge bases. For the

general knowledge bases used in Naive RAG and Iter-RetGen, the retriever is configured to fetch up to 16 knowledge chunks, applying a retrieval score threshold of $0.2$. For the general knowledge bases used in Self-Ask w/ Retrieval and IRCoT, where the retrieval chunks are used for a single follow-up question answering or the generation of single continuous rationale sentence, the reference chunks for whole rationale or final question answering are accumulated. The system retrieves 4 relevant chunks per request, maintaining the same score threshold of $0.2$. In the case of atomic knowledge bases, the retriever is set to retrieve 4 relevant atomic questions for each atomic query but with a higher threshold $0.5$ due to the shorter content length.

### A.3   DETAILED EXPERIMENTAL RESULTS ON OPEN-DOMAIN DATASETS

In addition to the methods outlined in Table 1, we also conduct experiments with a Knowledge Graph-based method, GraphRAG (Edge et al., 2024), to determine its effectiveness in multihop reasoning tasks. GraphRAG was inferred in both local and global modes. While we acknowledge the existence of other knowledge graph-based methods designed to tackle multihop questions-such as KGP (Wang et al., 2023b), which necessitates further fine-tuning, and ToG (Sun et al., 2024), which depends on external, predefined knowledge graphs-we contend that the additional requirements of these approaches render them not directly comparable to our approach, which is both training-free and relying only on a specific knowledge base.

The methods evaluated in this study are listed as follows:

- **Zero-Shot CoT**: Questions are addressed using solely the Chain-Of-Thought (CoT) technique, which prompts the LLMs to articulate its reasoning process step-by-step without the aid of example demonstrations or supplemental context. This method assesses the LLMs' intrinsic knowledge and reasoning capabilities in a zero-shot setting.

- **Naive RAG**: This approach employs dense retrieval from a flat knowledge base to procure relevant information for each question. The knowledge base consists of pre-embedded chunks are matched to the original question based on semantic similarity. The retrieval process is direct, without any intermediate task decomposition.

- **Self-Ask w/ Retrieval**: This method employs a task decomposition strategy wherein the LLMs is prompted to iteratively generate and answer follow-up questions, thereby breaking down complex problems into more manageable sub-tasks. General demonstrations illustrating the logic and methodology of task decomposition are provided for all benchmarks to guide the LLMs' reasoning process. Different to the original setting(Press et al., 2023), where the framework relies solely on LLM's own knowledge to answer each follow-up question, in this setting, we introduces an additional retrieval component. Relevant chunks are retrieved with the follow-up question as the query from a flat knowledge base to provide a reference context. What's more, we also limit the decomposition process to raise up to $N$ follow-up questions to align with other methods.

- **IRCoT**: This approach iteratively prompts LLMs to generate one more sentence of rationale with retrieved passages, and retrieves new passages with the newly generated reason. The original setting limit the process with a maximum token number(Trivedi et al., 2023). In our experiments, we limit the total number of iterations to the constant $N$ we used for our methods.

- **Iter-RetGen**: This method iteratively answers questions with retrieved passages, and uses the newly generated rationale and answer for the next-round retrieval. In this setting, we also limit the total number of iterations to the same $N$.

- **GraphRAG Local**: The knowledge base is pre-processed to construct a knowledge graph in accordance with the public guidance. The evaluation is inferred in local mode.

- **GraphRAG Global**: The knowledge base is pre-processed to construct a knowledge graph in accordance with the public guidance. The evaluation is inferred in global mode.

- **KAR$^3$ (Ours)**: The proposed knowledge-aware decomposition method iteratively decomposes complex questions into sub-questions and retrieves relevant knowledge up to a maximum of $N$ iterations. This process limits the context for the final answer to the five most useful knowledge chunks.

Table 4: Detailed performance comparison on multihop QA datasets. Best in bold, second-best underlined.

| Method | EM | F1 | Acc | Precision | Recall |
|---|---|---|---|---|---|
| Zero-Shot CoT | 32.60 | 43.94 | 53.60 | 46.56 | 43.97 |
| Naive RAG | 56.80 | 72.67 | 82.60 | 74.52 | 74.86 |
| Self-Ask w/ Retrieval | 57.00 | 71.40 | 80.00 | 73.25 | 73.95 |
| IRCoT | 51.40 | 67.30 | 81.00 | 69.32 | 72.15 |
| Iter-RetGen | 59.60 | 75.27 | 86.60 | 77.18 | 77.62 |
| GraphRAG Local | 0.00 | 10.66 | **89.00** | 5.90 | **83.07** |
| GraphRAG Global | 0.00 | 7.42 | 64.80 | 4.08 | 63.16 |
| KAR[3] (Ours) | **61.40** | **76.48** | 88.00 | **78.53** | 78.96 |

(a) HotPotQA

| Method | EM | F1 | Acc | Precision | Recall |
|---|---|---|---|---|---|
| Zero-Shot CoT | 35.67 | 41.40 | 43.87 | 41.43 | 43.11 |
| Naive RAG | 51.20 | 59.74 | 62.80 | 59.06 | 62.30 |
| Self-Ask w/ Retrieval | 60.60 | 69.06 | 75.00 | 67.88 | 73.15 |
| IRCoT | 55.00 | 63.83 | 70.40 | 62.47 | 68.86 |
| Iter-RetGen | 57.80 | 67.21 | 73.60 | 66.10 | 71.09 |
| GraphRAG Local | 0.00 | 11.83 | 71.20 | 6.74 | 75.17 |
| GraphRAG Global | 0.00 | 7.35 | 45.00 | 4.09 | 55.43 |
| KAR[3] (Ours) | **65.80** | **75.00** | **82.20** | **73.63** | **79.08** |

(b) 2WikiMultiHopQA

| Method | EM | F1 | Acc | Precision | Recall |
|---|---|---|---|---|---|
| Zero-Shot CoT | 12.93 | 22.90 | 23.47 | 24.40 | 24.10 |
| Naive RAG | 32.00 | 43.31 | 44.40 | 44.42 | 47.29 |
| Self-Ask w/ Retrieval | 38.20 | 46.76 | 51.40 | 46.75 | 51.00 |
| IRCoT | 36.00 | 47.57 | 49.20 | 48.70 | 50.30 |
| Iter-RetGen | 40.20 | 52.48 | 55.60 | 53.51 | 56.45 |
| GraphRAG Local | 0.60 | 9.62 | 49.80 | 5.73 | 55.82 |
| GraphRAG Global | 0.00 | 5.16 | 44.60 | 2.82 | 52.19 |
| KAR[3] (Ours) | **47.40** | **57.86** | **62.60** | **58.52** | **61.37** |

(c) MuSiQue

What's more, three more metrics are employed in Appendix. **Exact Match (EM)**, which assesses whether the response is identical to a predefined correct answer is applied as the community usually did. Furthermore, we encounter situations where a method achieves high accuracy (Acc) scores yet registers low F1 scores. To elucidate the underlying factors of such discrepancies, we also report on the **Recall** and **Precision** of the generated responses. Recall measures the proportion of relevant tokens from the answer labels that are captured in the response, while precision evaluates the relevance of the tokens in the generated answer with respect to the correct labels.

The detailed evaluation results across HotpotQA, 2WikiMultiHopQA, and MuSiQue are presented in Table 4. Notably, knowledge graph-based method, GraphRAG Local, excels in HotpotQA—a dataset predominantly comprised of 2-hop questions. However, in the other two datasets, which contain questions involving more hops, GraphRAG Local is merely on par with IRCoT. This highlights the challenge that knowledge graph-based methods face in addressing complex multihop questions.

Regarding GraphRAG, originally designed for the query-focused summarization (QFS) task as outlined by (Edge et al., 2024), we observe its suboptimal performance in both local and global modes compared to our method. GraphRAG exhibits a curious trend: it achieves higher accuracy and recall scores while performing lower on EM, F1, and Precision metrics. A closer analysis of GraphRAG's outputs reveals a tendency to echo the query and include meta-information about the answer within

Table 5: An Example of GraphRAG Local output on a HotpotQA question. The table showcases the tendency to repeat the question and include meta-information in its response.

| Question | Which country is home to Alsa Mall and Spencer Plaza? |
|---|---|
| Answer Labels | India |
| Answer of GraphRAG | Alsa Mall and Spencer Plaza are both located in Chennai, India [Data: India and Chennai Community (2391); Entities (4901, 4904); Relationships (9479, 1687, 5215, 5217)]. |

Table 6: Ablation study on hyper-parameter $N$. Recall$^*$ indicates the recall of supporting facts.

| $N$ | HotpotQA | | | 2Wiki | | | MuSiQue | | |
|---|---|---|---|---|---|---|---|---|---|
| | Recall$^*$ | F1 | Acc | Recall$^*$ | F1 | Acc | Recall$^*$ | F1 | Acc |
| 1 | 42.96 | 59.46 | 70.20 | 40.41 | 41.08 | 43.00 | 31.20 | 32.55 | 32.80 |
| 2 | 82.04 | 74.27 | 84.80 | 78.83 | 70.22 | 77.20 | 56.43 | 48.46 | 50.00 |
| 3 | 90.16 | 76.90 | 87.20 | 87.71 | 72.84 | 79.40 | 64.82 | 53.50 | 57.20 |
| 4 | 92.46 | 76.49 | 87.80 | 92.86 | 74.68 | 81.80 | 69.87 | 55.73 | 59.40 |
| 5 | 92.83 | 76.48 | 88.00 | 94.06 | 75.00 | 82.20 | 73.08 | 57.86 | 62.60 |
| 6 | 93.35 | 77.67 | 89.00 | 94.76 | 75.12 | 81.80 | 74.88 | 57.03 | 61.20 |
| 7 | 93.68 | 77.32 | 88.80 | 94.91 | 75.44 | 82.40 | 76.07 | 56.66 | 61.40 |
| 8 | 93.78 | 76.88 | 88.40 | 95.06 | 75.16 | 82.00 | 76.72 | 57.65 | 62.40 |
| 9 | 93.78 | 76.99 | 88.60 | 95.11 | 74.89 | 81.80 | 76.90 | 57.17 | 61.40 |
| 10 | 93.78 | 77.52 | 89.00 | 95.16 | 75.09 | 82.00 | 77.20 | 57.69 | 62.40 |

its graph structure. Despite attempts to refine its QA prompt, this behavior persists. An illustrative example is presented in Table 5, which shows GraphRAG Local's response to a question from HotpotQA.

Table 6 lists the granular performance metrics according to those we shown in Figure 3 for the ablation study on the iteration upper bound $N$. Different to the **Recall** we reported in Table 4, which indicates the recall tokens of the answer labels, the **Recall**$^*$ here represents the recall of the supporting facts provided by these datasets.

As introduced in the limitation discussion section, we have carried out a series of experiments utilizing GPT-3.5. The outcomes of these experiments are delineated in Table 7. For these specific trials, we substituted GPT-4 (1106-Preview) with GPT-3.5 (1106-Preview) as the language model, while maintaining all other experimental settings identical to those employed in the experiments summarized in Table 1.

A.4 Evaluation on Legal Benchmarks

In this subsection, we present the performance of our approach on two legal benchmarks: Law-Bench Fei et al. (2023) and Open Australian Legal QA Butler (2023). Before doing so, we provide a brief description of each benchmark.

**LawBench** LawBench is a comprehensive legal benchmark for Chinese laws. It comprises 20 meticulously designed tasks aimed at accurately assessing the legal capabilities of LLMs. Unlike some existing benchmarks that rely solely on multiple-choice questions, LawBench includes a variety of task types that are closely related to real-world applications. These tasks encompass legal entity recognition, reading comprehension, crime amount calculation, and legal consulting, among others. Since not all tasks are RAG-oriented (e.g., reading comprehension), we have selected 6 specific tasks, which are detailed in Table 8. The number of questions of each task is 500.

We also provide example questions of these tasks for the readers reference (translated using GPT-4).

```
1-1: Answer the following question by directly providing the content of
    ↪ the article:What is the content of Article 76 of the Securities Law
    ↪ ?
```

Table 7: Performance comparison of implementations with *GPT-3.5*. Best Accuracy in bold, second-best Accuracy underlined.

| Method | HotpotQA | | 2Wiki | | MuSiQue | |
|---|---|---|---|---|---|---|
| | F1 | Acc | F1 | Acc | F1 | Acc |
| Self-Ask w/ Retrieval | 49.52 | 61.40 | 53.83 | **60.00** | 31.05 | 35.20 |
| IRCoT | 56.39 | 68.40 | 40.31 | 46.00 | 33.93 | 34.40 |
| Iter-RetGen | 48.63 | 66.80 | 44.32 | 55.20 | 25.77 | 37.80 |
| KAR[3] (**Ours**) | 46.37 | **68.80** | 41.95 | 58.20 | 26.80 | **39.60** |

Table 8: Overview of LawBench tasks

| Task No. | Task | Type | Metric |
|---|---|---|---|
| 1-1 | Statute Recitation | Generation | F1 |
| 1-2 | Legal Knowledge Q&A | Single Choice | EM |
| 3-1 | Statute Prediction (Fact-based) | Multiple Choices | EM |
| 3-2 | Statute Prediction (Scenario-based) | Generation | F1 |
| 3-6 | Case Analysis | Single Choice | EM |
| 3-8 | Consultation | Generation | F1 |

```
1-2: According to the 'Securities Law', which of the following statements
    ↪ about stock exchanges is incorrect? A: Without the permission of
    ↪ the stock exchange, no entity or individual may publish real-time
    ↪ securities trading information; B: The stock exchange may restrict
    ↪ trading on securities accounts that exhibit major abnormal trading
    ↪ conditions as needed, and report to the securities regulatory
    ↪ authority under the State Council for record; C: The accumulated
    ↪ property of a member-based stock exchange belongs to the members,
    ↪ and their rights are jointly enjoyed by the members; during its
    ↪ existence, the accumulated property may not be distributed to the
    ↪ members; D: The stock exchange formulates listing rules, trading
    ↪ rules, member management rules, and other relevant rules in
    ↪ accordance with securities laws and administrative regulations, and
    ↪  reports to the securities regulatory authority under the State
    ↪ Council for record.
3-1: Based on the following facts and charges, provide the relevant
    ↪ articles of the Criminal Law. Facts: The Yushu City, Jilin Province
    ↪ , accused that on November 15, 2015, the defendant He signed a car
    ↪ rental agreement with Guo, the owner of a taxi with license plate
    ↪ number . The agreement stipulated a monthly rent of RMB 3,900.00,
    ↪ payable monthly. On January 19, 2016, without the knowledge of Guo,
    ↪  the defendant He concealed the truth and falsely claimed to be the
    ↪  owner of the taxi. He signed a car rental agreement with the
    ↪ victim Ma, with a monthly rent of RMB 3,800.00 and a rental period
    ↪ of one year, collecting a total of RMB 50,600.00 from Ma for one
    ↪ year's rent and vehicle deposit. On February 26, 2016, the taxi was
    ↪  retrieved by its owner Guo from the victim Ma. The victim Ma
    ↪ repeatedly asked the defendant He to return the rent and deposit,
    ↪ but the defendant He refused to return them. The prosecution
    ↪ provided evidence including the defendants confession, the victims
    ↪ statement, witness testimonies, and documentary evidence, and
    ↪ believed that the defendant He, with the purpose of illegal
    ↪ possession, defrauded others of their property by fabricating facts
    ↪  and concealing the truth during the signing and performance of the
    ↪  contract. The amount was relatively large, and his actions
    ↪ violated the provisions of Article  of the Criminal Law of the
    ↪ Peoples Republic of China, and he should be held criminally
    ↪ responsible for . Charge: Contract Fraud.
3-2: Please provide the legal basis according to the specific scenario
    ↪ and question, only the content of the specific legal provision is
    ↪ needed, each scenario involves only one legal provision. Scenario:
```

Table 9: Evaluation Results on Legal Benchmarks (Metric is **F1 / EM** as indicated in Table 8)

| Task | | Zero-Shot CoT | GraphRAG Local | Ours (N=5) |
|---|---|---|---|---|
| LawBench | 1-1 | 21.31 | 23.27 | **78.58** |
| | 1-2 | 54.24 | 62.60 | **70.60** |
| | 3-1 | 53.32 | 74.60 | **83.16** |
| | 3-2 | 27.51 | 25.98 | **46.05** |
| | 3-6 | 51.16 | 47.64 | **61.91** |
| | 3-8 | 17.44 | 18.43 | **23.58** |
| Open Australian Legal QA | | 25.10 | 34.35 | **63.34** |

Table 10: Evaluation Results on Legal Benchmarks (Metric is **Acc**)

| Task | | Zero-Shot CoT | GraphRAG Local | Ours (N=5) |
|---|---|---|---|---|
| LawBench | 1-1 | 1.23 | 16.60 | **90.12** |
| | 1-2 | 54.00 | 63.40 | **70.60** |
| | 3-1 | 49.90 | 75.40 | **88.82** |
| | 3-2 | 15.83 | 27.60 | **67.54** |
| | 3-6 | 51.12 | 57.00 | **62.73** |
| | 3-8 | 49.70 | 58.80 | **61.72** |
| Open Australian Legal QA | | 16.48 | 88.27 | **98.59** |

```
    ↪ A cargo ship arrives at the port of discharge, but the consignee
    ↪ fails to arrive in time to collect the goods. Under which legal
    ↪ provision can the captain unload the goods at another appropriate
    ↪ place?
3-6: One year after the bar opened, the business environment changed
    ↪ drastically, and all partners held a meeting to discuss
    ↪ countermeasures. According to the 'Partnership Enterprise Law,' the
    ↪  following voting matters are considered valid votes: A: Zhang
    ↪ believes that the name 'Tongcheng' is not attractive and proposes
    ↪ to change it to 'Tongsheng Bar.' Wang and Zhao agree, but Li
    ↪ opposes; B: In view of the sluggish business, Wang proposes to
    ↪ suspend operations for one month for renovation and reorganization.
    ↪  Zhang and Zhao agree, but Li opposes; C: Due to the urgent needs
    ↪ of the bar, Zhao proposes to sell a batch of coffee machines to the
    ↪  bar. Zhang and Wang agree, but Li opposes; D: Given the four
    ↪ partners' lack of experience in bar management, Li proposes to
    ↪ appoint his friend Wang as the managing partner. Zhang and Wang
    ↪ agree, but Zhao opposes.
3-8: Resident A rented out the house to B. With A's consent, B renovated
    ↪ the rented house and sublet it to C. C unilaterally altered the
    ↪ load-bearing structure of the house. Why can A request B to bear
    ↪ liability for breach of contract?
```

**Open Australian Legal QA**  The benchmark consists of 2,124 questions and answers synthesized by GPT-4 from the Australian legal corpus. All questions are of the generation type. One example is: "What is the landlord's general obligation under section 63 of the Act in the case of Anderson v Armitage [2014] NSWCATCD 157 in New South Wales?"

Evaluation results are listed in Table 9, where we only compare to "GraphRAG Local", as it generally performs better than "GraphRAG Global" on these tasks.

For the aforementioned reasons, we also use GPT-4 to evaluate all experimental results, reporting the accuracy (**Acc**) in Table 10. When comparing the results in Table 9 and Table 10, we observe that the order of the results is preserved, even though some metrics change significantly. In the following section, we aim to identify the reasons behind these changes, which may provide valuable insights for designing better metrics to evaluate RAG frameworks in the future.

1. The accuracy of our approach increases significantly for generation tasks (1-1, 3-2, Open Australian Legal QA). For these tasks, our answers are often semantically equivalent but syntactically different from the golden answers. This explains the improved metric performance, as GPT-4 can compare the semantic content of the answers. This also applies to the "GraphRAG Local" results for the "Open Australian Legal QA" task.

2. The accuracy of "GraphRAG Local" decreases for generation tasks 1-1 and 3-2. These tasks involve statute recitation and prediction, requiring the retrieval of specific articles. Upon detailed examination, We find that "GraphRAG Local" often fails to retrieve the correct articles or references the wrong ones, but it tends to repeat the legal information. Therefore, token-level recall can be improved by simply rephrasing legal names and common prefixes, such as "According to XX law, XX articles...".

3. Both our approach and "GraphRAG Local" show significant accuracy improvements on task 3-8. Besides the reason mentioned in the first point, the quality of the golden answers may also contribute to this difference. The questions and golden answers in task 3-8 are sourced from a consulting website, resulting in varying quality. For example, one question asks "Do the children from the original marriage have an obligation to support the father?" However, the provided golden answer includes an irrelevant article, "Article 1067," which pertains to parents' obligations to support minor children.

```
Question: In the case where both parents are divorced and have
    ↪ formed their own families with new children, and according
    ↪ to the court's judgment, the father is required to pay
    ↪ monthly child support to the mother until the child is 18
    ↪ years old. Do the children from the original marriage have
    ↪ an obligation to support the father?
Reference Answer: In our country, biological children have an
    ↪ obligation to support their parents who are divorced. The
    ↪ relationship between children and parents does not dissolve
    ↪ because of the divorce of the children or parents.
    ↪ Supporting parents is a legal obligation of the children. If
    ↪  the children do not support the elderly, the parents can
    ↪ directly file a lawsuit in the People's Court requesting the
    ↪  court to order the daughter to pay support. Legal basis:
    ↪ Article 1067 of the Civil Code of the People's Republic of
    ↪ China states that if parents do not fulfill their obligation
    ↪  to support, minor children or adult children who cannot
    ↪ live independently have the right to request support from
    ↪ their parents. If adult children do not fulfill their
    ↪ obligation to support, parents who are incapacitated or have
    ↪  difficulties in living have the right to request support
    ↪ from adult children. Article 1084 states that the
    ↪ relationship between parents and children does not dissolve
    ↪ because of the parents' divorce. After the divorce, whether
    ↪ the children are directly raised by the father or the mother
    ↪ , they are still the children of both parents. Lawyer's
    ↪ explanation: After the parents are divorced, adult children
    ↪ still have the obligation to support both parents. The
    ↪ obligation of adult children to support their parents does
    ↪ not change because of changes in the parents' relationship.
    ↪ If adult children do not fulfill their obligation to support
    ↪ , parents who are incapacitated or have difficulties in
    ↪ living have the right to request support from adult children
    ↪ .
Our Answer: Yes, there is an obligation to support. Legal basis:
    ↪ According to Article 1069 of the Civil Code of the People's
    ↪ Republic of China, it is stipulated that: 'The obligation of
    ↪  children to support their parents does not terminate
    ↪ because of changes in the parents' marital relationship.'
    ↪ Therefore, even if the parents are divorced and remarried
    ↪ with new children, the original children still have an
    ↪ obligation to support their parents.
```

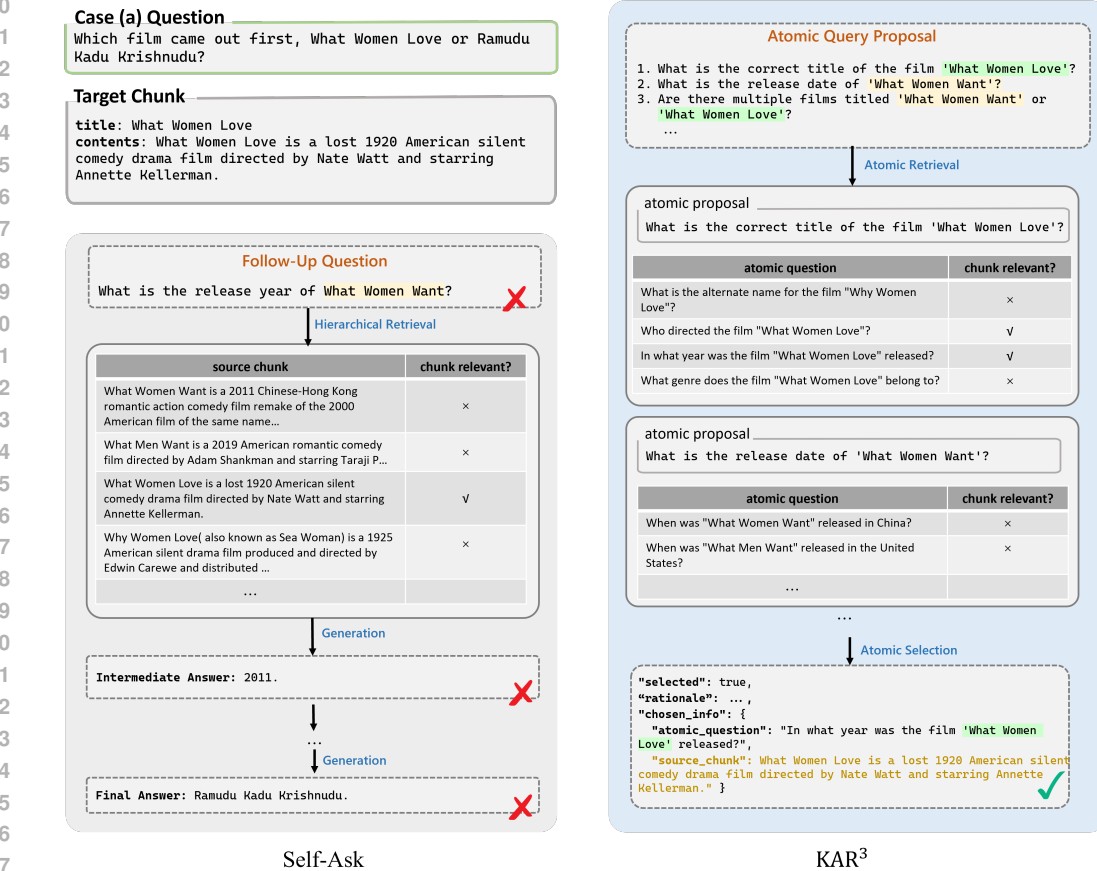

Self-Ask          KAR³

Figure 4: Case (a): Given the lesser-known film "What Women Love" as opposed to the more popular "What Women Want," single-path methods like Self-Ask on the left are predisposed to generating follow-up questions about the latter, leading to an incorrect final answer. Conversely, KAR³ can effectively discern the intended meaning of the original question by positing several atomic queries and postpone the task understanding to atomic selection phase with relevant atomic questions provided, and subsequently arriving at an accurate conclusion.

4. The accuracy of all methods on choice tasks 1-2, 3-1, and 3-6 almost coincides with the F1 score, as expected. An exception is task 3-1, where the difference is mainly due to GPT-4's capacity to understand Chinese, particularly in distinguishing numbers in Arabic and Chinese. In Chinese law, all numbers are written in Chinese, while in the golden answers, all numbers are given in Arabic.

## A.5 REAL CASE STUDIES

This section presents three case studies from our evaluation benchmark to illustrate the underlying principles of our proposed decomposition pipeline, as detailed in Algorithm 1. Through these real-world examples, we aim to highlight the benefits of our systematic approach. These cases will shed light on how each step of the pipeline contributes to improved performance and the insights gained from their implementation.

Our task decomposition strategy involves generating multiple atomic queries rather than producing a single deterministic follow-up question, as demonstrated in the Self-Ask approach. Contemporary decomposition methods typically employ a generative model to formulate a singular follow-up question. However, this approach carries an intrinsic risk of generating erroneous questions, potentially leading to an incorrect decomposition pathway and, ultimately, an erroneous answer. Consider the Case (a) depicted in Figure 4, where the original question pertains to a film titled "What Women

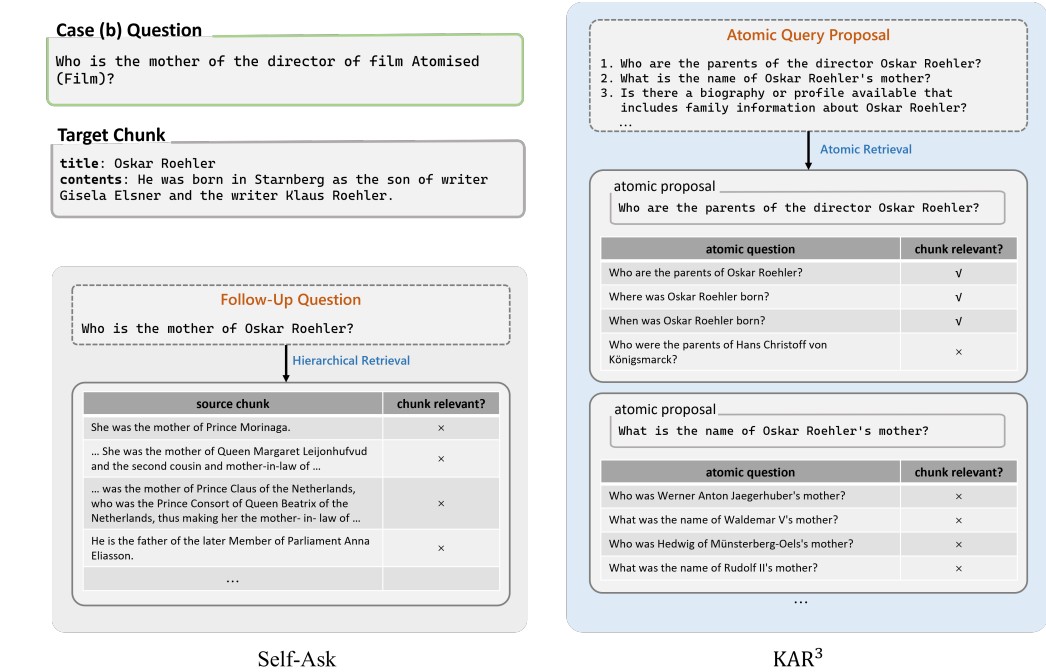

Figure 5: Case (b): By proposing multiple atomic queries, KAR³ effectively retrieves the relevant knowledge chunk, whereas the single deterministic follow-up question approach employed by Self-Ask fails to align with the knowledge base's schema, resulting in a retrieval failure.

Love." Due to the existence of a more prominent film, "What Women Want," the employed language model tends to 'correct' the original question. Consequently, methods like Self-Ask (as shown on the left side of Figure 4) generate only one follow-up question related to this erroneously assumed object. In the illustrated instance, although the target chunk has been retrieved due to the similarity in embeddings, a 'false' intermediate answer is produced for the 'false' follow-up question, culminating in an incorrect final response. In contrast, our methodology posits atomic queries concerning both "What Women Love" and "What Women Want," thereby seeking to clarify the true intent of the initial question. With both films existing and relevant atomic questions being retrieved, our approach subsequently gains the advantage of verifying the question's intent and selecting the correct and most pertinent chunk during the atomic selection phase.

Furthermore, the discrepancy between the formulation of the corpus and the query, is another critical factor advocating for a multi-query approach over a singular deterministic one. The presentation gap can impede the retrieval process even when the generated follow-up question is semantically accurate. For instance, as illustrated in Case (b) in Figure 5, a single-path method such as Self-Ask on the left side might directly inquire 'Who is the mother of Oskar Roehler?' However, the knowledge base articulates familial relationships using a different schema, 'A is the son of B and C' in this case, thus the retrieval process falters despite the correctness of the question. Even when we applied the hierarchical retrieval to Self-Ask, the Self-Ask with Hierarchical Retrieval did not succeed in bridging this gap. In contrast, our approach, which generates multiple atomic queries, encompasses a broader range of phrasings that correspond to the diverse representations in the knowledge base. In the depicted case, while the atomic query specifically asking for Oskar Roehler's mother encounters the same retrieval issue, an alternative query seeking information about his parents successfully retrieves the target chunk. This exemplifies how our method's flexibility in query generation enhances the likelihood of aligning with the knowledge base's structure and obtaining accurate information.

Our methodology emphasizes the retrieval of atomic questions rather than directly retrieving chunks. This design choice is exemplified in Case (b) depicted in Figure 5. The knowledge chunk in the corpus is structured using the pattern 'A ... as the son of B and C', which poses challenges for direct retrieval by queries such as 'Who is the mother of ...'. In our specialized knowledge base, such direct queries tend to retrieve chunks conforming to the patterns 'A is the mother of B' or 'A is the father

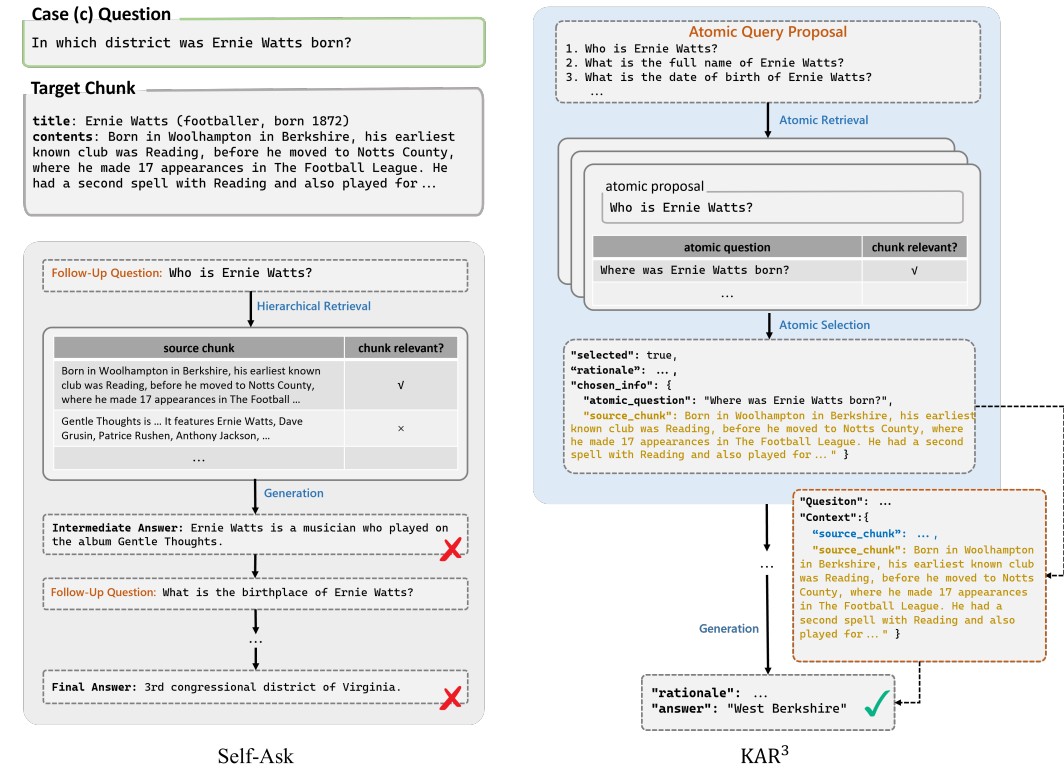

Figure 6: Case (c): KAR³ has the advantage of leveraging a concise list of atomic questions for targeted selection and retaining full chunks for rich contextual support. Conversely, Self-Ask's approach, although successful in retrieving relevant chunks, is compromised by its dependency on intermediate answers for context, which ultimately results in the generation of incorrect final answers.

of B'. By utilizing atomic questions as intermediaries for retrieval, our approach effectively narrows the gap between a single query and the multiple sentence structures found in the knowledge base. It facilitates bridging the expression pattern differences exemplified by 'the mother of' versus 'the son of' in this scenario.

In contrast to methods like Self-Ask, which only retains intermediate answers for subsequent processing, our method preserves the entire chunk as contextual information. During the atomic selection phase, we present a list of atomic questions as candidate summaries of the relevant content from the original chunk. This strategy significantly reduces token usage and simplifies the process of selecting the pertinent information. Case (c) in Figure 6 demonstrates the dual benefits of our approach: first, by selecting from a curated list of atomic questions, we streamline the identification of relevant information; second, by retaining the entire selected chunk rather than just the intermediate answer, we ensure a rich context is maintained for accurate and comprehensive subsequent processing. While the Self-Ask method on the left retrieves the target chunk, it fails to correctly identify the pertinent 'Ernie Watts' due to the excessive contextual information. Since retrieved chunks in Self-Ask are discarded after generating an intermediate answer, the method potentially follows an incorrect pathway, leading to an inaccurate conclusion. In contrast, our approach can efficiently filter and select the appropriate atomic question from a concise list. Although the atomic question in this round pertains to the role of Ernie Watts, there is no need to inquire further about his birthplace, as this information is encapsulated within the selected chunk, which remains available for context in subsequent rounds.

Table 11: Performance comparison: Zero-Shot vs. Few-Shot.

| Method | HotpotQA | | 2Wiki | | MuSiQue | |
|---|---|---|---|---|---|---|
| | F1 | Acc | F1 | Acc | F1 | Acc |
| Zero-Shot Self-Ask w/ Retrieval | 55.76 | 76.20 | 54.98 | 76.20 | 40.97 | 50.40 |
| Self-Ask w/ Retrieval | 71.40 | 80.00 | 69.06 | 75.00 | 46.76 | 51.40 |
| Zero-Shot IRCoT | 58.22 | 75.80 | 49.69 | 60.20 | 37.17 | 43.00 |
| IRCoT | 67.30 | 81.00 | 63.83 | 70.40 | 47.57 | 49.20 |

## A.6 PROMPT DESIGN

Our approach employs four distinct prompts, detailed at the end of the appendix. (1) Atomic question tagging prompt: the one used to pre-processing the source paragraphs that linking each paragraphs with several atomic questions; (2) Atomic query proposer prompt: the one used when generating multiple atomic query proposals, referring to line 3 in Algorithm 1; (3) Atomic question selection prompt: the one used when selecting the most useful atomic question from the given question list, referring to line 5 in Algorithm 1; (4) Question answering prompt: the one applied upon exiting the decomposition loop to generate the final answer to the given question, as described in line 14 of Algorithm 1.

**Demonstration Discussion** In our current experiments, all prompts are zero-shot, meaning no demonstrations are provided to illustrate the expected reasoning logic. To explore whether demonstrations could enhance performance, we designed an ablation study. We adapted the Self-Ask w/ Retrieval and IRCoT methodologies previously employed, modifying the prompts and task descriptions to create zero-shot, demonstration-free variants of these methods. These were denoted as **Zero-Shot Self-Ask w/ Retrieval** and **Zero-Shot IRCoT**. The results of the experiment are presented in Table 11.

The experimental results reveal that the Zero-Shot Self-Ask w/ Retrieval method experiences a marginal decline in accuracy for the 2WikiMultiHopQA and MuSiQue datasets, potentially due to the inherent randomness in generation. However, the inclusion of demonstrations significantly improves all F1 scores and enhances the overall performance of the IRCoT method. This suggests that demonstrations could be particularly beneficial for methods that rely on a step-by-step decomposition approach. Consequently, integrating demonstrations is identified as a promising direction for future work within the $KAR^3$ framework.

**Atomic Question Tagging Prompt**

```
# Task
Your task is to extract as many questions as possible that are relevant
and can be answered by the given content.  Please try to be diverse
and avoid extracting duplicated or similar questions.  Make sure your
question contain necessary entity names and avoid to use pronouns like
it, he, she, they, the company, the person etc.

# Output Format
Output your answers line by line, with each question on a new line,
without itemized symbols or numbers.

# Content
{content}

# Output
```

**Atomic Query Proposer Prompt**

```
# Task
Your task is to analyse the providing context then raise atomic
sub-questions for the knowledge that can help you answer the question
better.  Think in different ways and raise as many diverse questions as
possible.

# Output Format
Please output in following JSON format:
{{
    "thinking":  <your thinking for this task, including analysis to
the question and the given context>,
    "sub_questions":  <a list of sub-questions indicating what you
need>
}}

# Context
The context we already have:
{chosen_content}

# Question
{content}

# Your Output
```

**Atomic Question Selection Prompt**

```
# Task
Your task is to analyse the providing context then decide which
sub-questions may be useful to be answered before you can answer
the given question.  Select a most relevant sub-question from the
given question list, avoid selecting sub-question that can already
be answered with the given context or with your own knowledge.

# Output Format
Please output in following JSON format:
{{
    "thinking":  <your thinking for this selection task>,
    "question_idx":  <a sub-question index, an integer from 1 to
{num_atom_questions}>
}}

# Context
The context we already have:
{chosen_content}

# Sub-Questions You Can Choose From
{atom_question_list_str}

# Question
{content}

# Your Output
```

**Question Answering Prompt**

```
# Task
Your task is to answer a question referring to a given context, if
any.  For answering the Question at the end, you need to first read the
articles, reports, or context provided, then give your final answer.

# Output format
Your output should strictly follow the format below.  Make sure your
output parsable by json in Python.
{{
    "answer":  <Your Answer, format it as a string.>,
    "rationale":  <rationale behind your choice>
}}

# Context, if any
{context_if_any}

# Question
{content}{yes_or_no_limit}

Let's think step by step.
```

