# OpenReview forum: "From Complex to Atomic: Enhancing Augmented Generation via Knowledge-Aware Dual Rewriting and Reasoning"
_ICLR.cc/2025/Conference — Submitted to ICLR 2025_

### Official Review · Reviewer_vTwe · 2024-11-03

**Soundness:** 2
**Presentation:** 2
**Contribution:** 2
**Rating:** 3
**Confidence:** 4

**Summary:**

This paper points out that the semantic retrieval is inadequate for mining deep, domain-specific knowledge for performing logical reasoning. The authors propose a knowledge-aware task decomposition strategy that extracts, comprehends, and utilizes atomic domain knowledge while constructing a coherent rationale.

**Strengths:**

S1.  Supplementing semantic retrieval-based RAG with fine-grained and atomic knowledge is a novel and realistic practice.

S2. The proposed strategy is easy to understand and follow.

S3. The experimental results are explicitly competitive.

**Weaknesses:**

W1. Regardless of the novel motivation, the explanation of the motivation is poor. The authors claim that ‘Current RAG methods relying on semantic retrieval fall short in deep domain-specific reasoning’. However, they do not specify the main challenges in domain-specific reasoning and why current RAG methods fall short in this setting. It will make this paper clearer and more convincing to clarify the above points.

W2. The proposed strategy includes using GPT4 to atomize raw chunks into sub-queries, using GPT4 to decompose the incoming queries, and using GPT4 to select optimal contexts and judge the termination. It’s more like a pipeline. Please point out the main technical challenges through this pipeline and the main technical contributions to solve the challenges.

W3. The proposed strategy relies highly on calling LLMs, which costs much. The dependence on the capability of GPT4 makes the proposed strategy inefficient and cost-intensive. Is there any cost analysis comparing the proposed strategy to baselines?

**Questions:**

The proposed strategy is dependent on GPT-4, which is expensive. Do cheaper LLMs fit in the pipeline? Are there any experiments or analyses with less expensive models? How might the performance be expected to change with different LLMs?

---

> ### Author Response · Authors · 2024-11-24
> **Author Response to Reviewer vTwe (part 1)**
>
> ### *Q1.The main challenges in domain-specific reasoning and why current RAG methods fall short in this setting.*
>
> In our work, domain-specific knowledge pertains to information extracted from the corpus designated for RAG, which may not be fully covered by the pre-trained LLMs and can sometimes diverge from LLMs' intrinsic knowledge. For instance, within a particular domain, "ice cream" might be used as a unique project name rather than referencing the typical frozen dessert. This domain-specific knowledge serves as external knowledge to assist LLMs in generating responses that are consistent with the knowledge from the RAG corpus. **Nevertheless, integrating domain-specific knowledge to domain-aligned generation introduces significant challenges, which include the difficulties of a) effectively extracting domain-specific knowledge, b) precisely retrieving it via queries, and c) successfully constructing coherent reasoning chain based on it. To tackle these challenges, we propose a novel approach encompassing knowledge atomization, atomic pair retrieval, and query decomposition with reference context.** Although current RAG approaches such as Self-Ask and SeaChain similarly employ LLMs to sequentially generate sub-questions, they fall short in adequately harnessing the knowledge contained within RAG data. They also fail to comprehensively tackle all three of the aforementioned challenges.
>
> To illustrate the limitation of current RAG, Table 4 presents an example. The regular RAG systems tend to rely on common knowledge to decompose query into subqueries, often overlooking the availability of corresponding answers in the RAG data, as shown in the **part one** of example analysis of the response to CQ3. Our method, illustrated in the **part two** of example analysis, incorporates the reference context from previously retrieved data to guide the subsequent query decomposition. This approach generates potential reasoning paths based on the domain-specific knowledge from RAG data, ultimately arriving at the correct answer with a complete reasoning chain.
>
> In the revised version, we will highlight the challenges in domain-specific reasoning and provide detailed explanation of on how current RAG methods fall short in this setting.
>
> ### *Q2. Main technical challenges besides the proposed pipeline.*
>
> As referred in the response to Q1, the integration of domain-specific knowledge into domain-aligned generation is fraught with challenges. The challenges lie in a) effectively extracting domain-specific knowledge, b) precisely retrieving relevant knowledge via queries, and c) successfully constructing coherent reasoning chain based on it. Current RAG approaches primarily address only a subset of the challenges. Designing effective individual components and optimizing them jointly within the RAG framework is a significant technical hurdle, aside from pipeline considerations.
>
> First, we propose a novel knowledge representation, atomic question, to exploit the multifaceted knowledge within RAG data.
> Secondly, we implement a query decomposition strategy to produce subqueries. This facilitates a more precise question-to-question retrieval process, thereby better aligning the query with the knowledge representation.
> Thirdly, we propose decomposing query to multiple subqueries with awareness of the relevant domain-specific knowledge, presented as reference contexts. Our design enables the exploration of potential reasoning paths through multiple subqueries and the selection of the most confident one to pursue. This design is more efficient than ProbTree that transverse each decomposition path and more robust than Self-Ask that relies on deterministic sequential decomposition steps.
>
> To clarify the contribution of each proposed component, we conduct ablation study in the main paper (line 483~521). More specially, we replace each proposed module with a simpler, standard module typically used in regular RAG systems. The results, presented in Table 5, demonstrate the effectiveness of each component. It is evident that each element is vital to our method's performance, particularly as the complexity of the questions increases, such as in the multi-hop scenarios seen with MuSiQue.
>
> Table 5. Ablation study on components of KAR³.
> | Variant Method               | Substitute Module   | HotpotQA F1 | HotpotQA Acc | 2Wiki F1 | 2Wiki Acc  | MuSiQue F1 | MuSiQue Acc |
> |------------------------------|---------------------|----------|------|-------|------|---------|------|
> | w/o Atomic Query Proposer    | Single Query        | 75.06    | 85.60| 70.19 | 76.40| 49.67   | 52.20|
> | w/o Atomic Pair Retriever    | Chunk Retriever    | 76.31    | 86.60| 67.14 | 72.40| 49.05   | 53.00|
> | w/o Atomic Selector          | Chunk Selector      | 72.80    | 83.20| 61.65 | 65.80| 49.31   | 53.40|
> | KAR³ **(Ours)**   | -                   | 76.48    | 88.00| 75.00 | 82.20| 57.86   | 62.60|

---

> ### Author Response · Authors · 2024-11-24
> **Author Response to Reviewer vTwe (part 2)**
>
> ### *Q3. Cost analysis and comparison with other baselines*
>
> To address reviewer's concern, we have conducted a comprehensive cost analysis to evaluate our model's API consumption compared to other baseline methods, and token consumption per QA of various methods are shown in Table 1. Our method consumes fewer tokens per QA compared to ProbTree and IRCoT, and is on par with Iter-RetGen. Notably, our approach significantly outperforms these baselines in both F1 and Accuracy metrics.
>
> Moreover, we implement our method and other baselines using the open-source LLM, meta-llama/Llama-3.1-70B-Instruct (including chunk atomization). The results are presented in Table 2, demonstrating that our method not only maintains excellent performance after replacing GPT-4 with an open-source LLM, but also significantly outperforms the other baselines. For a more detailed discussion, please refer to our response to CQ1.
>
> ### *Q4. Do cheaper LLMs fit in the pipeline?*
>
> We replace Gpt-4 with the open-source LLM, meta-llama/Llama-3.1-70B-Instruct (including chunk atomization). The results presented in Table 2, indicate that this implementation could maintains excellent performance after replacing GPT-4 with an open-source LLM. For a more detailed results, please refer to our response to CQ1.
> It should be noted that language models must possess an adequate capacity to follow instructions contained within prompts effectively. Consequently, language models that are too weak are not suitable for performing the complex reasoning tasks that our objectives demand.

---

> > ### Comment · Reviewer_vTwe · 2024-11-26
> >
> > Thank you for your reply. Some of my concerns are solved. Here are some additional questions:
> >
> > Q1: The cost analysis does not include the chunk atomization as it’s performed offline. However, chunk atomization is an extra procedure proposed by this paper. I am still curious about the detailed statistics, including the input/output token costs, the number of LLM calls, and the time cost.
> >
> > Q2: How many atomized questions are generated for each chunk? What’s the advantage over simply atomizing the chunks into sub-chunks? According to my understanding, the semantic space mismatch between the query and the chunks is mitigated by the embedding model used for retrieving. Are there any experimental results for this point? For example, a variant of KAR^3 atomizing chunks into plain texts instead of question?
> >
> > Q3: How do we establish the association between chunks with the atomic question as claimed in Line 283? It seems helpful to locate relevant atomic questions and link associated chunks.
> >
> > Q4: I am unsure about the atomic selector in Eq (6). Is it the same LLM used for atomization and then prompted to select useful atomic questions? What’s the cost of the selector?
> >
> > Q5: I observe that only the baseline IRCoT shows an improvement with LLaMA3 than with GPT4, what’s the reason behind the improvement?

---

> > > ### Author Response · Authors · 2024-11-28
> > > **Author Response to Reviewer vTwe (round 2, part1)**
> > >
> > > We appreciate the reviewers' careful consideration of our responses and are gratified to hear that our clarifications have resolved several of the concerns initially raised. We will provided detailed responses to the remaining questions and comments to ensure that all issues are properly addressed.
> > >
> > > ### *Q1.Detailed chunk atomization statistics.*
> > >
> > > We provide detailed statistics on chunk atomization in Table 6 for your reference. All chunks are derived from the context paragraphs of the sampled QA data, and their counts vary by dataset, as described in line 414 ~ 418 of the paper. Since each chunk is atomized by a LLM call, the number of LLM calls is equivalent to the chunk count, which is listed in the last column of Table 6. The input token size is primarily determined by the chunk size, while the output token size depends on the size of generated atomic questions. Additionally, the time required for each LLM call depends not only on the number of tokens but also on the workload of the endpoints used, making it an unstable metric. For example, the time cost for a single atomization call on HotpotQA range from 0.67s to 17.61s, according to our logs.
> > >
> > > Table 6. Token consumption (average/chunk) and chunk count statistics.
> > >
> > > | Dataset  | Prompt(input) | Completion(output) | Total    | Calls(chunk count) |
> > > |----------|---------------|--------------------|----------|--------------------|
> > > | HotpotQA |      209      |         129        |   338    |        4950        |
> > > |   2Wiki  |      199      |         122        |   321    |        3410        |
> > > |  MuSiQue |      197      |         123        |   320    |        7120        |
> > >
> > >
> > >
> > > ### *Q2.Advantage of atomizing over chunking.*
> > >
> > > ### *Q2-1. How many atomized questions are generated for each chunk?*
> > >
> > > We count the number of generated atomic questions and compute the average number per chunk for each dataset. The results are detailed in Table 7.
> > >
> > > Table 7. Atomic Question Count v.s.  Sub-chunks Count. (average/chunk).
> > >
> > > | Dataset   | Atomic Questions (**Ours**) | Sub-chunks (Plain Text) |
> > > |-----------|-----------------------------|------------------------|
> > > | HotpotQA  |          10.36              |          4.11          |
> > > |   2Wiki   |           9.45              |          3.20          |
> > > |  MuSiQue  |           9.62              |          3.62          |
> > >
> > > ### *Q2-2. What’s the advantage over simply atomizing the chunks into sub-chunks?*
> > >
> > > **When chunks are directly divided into sub-chunks, the semantic coherence between them may be compromised. This disruption can affect aspects, such as pronoun references, the chronological order indicated by temporal adverbs, and causal relationships.**
> > > For example, if the preceding two sentences are divided into two sub-chunks:
> > >
> > >     Sub-chunk 1: "When chunks are directly divided into sub-chunks, the semantic coherence between them may be compromised."
> > >     Sub-chunk 2: "This disruption can affect aspects, such as pronoun references, the chronological order indicated by temporal adverbs, and causal relationships."
> > >
> > > It becomes apparent that the term "disruption", in the second sub-chunk loses its connection to the concept of chunk division expressed in the first sub-chunk. Our proposed chunk atomization functions as a context-aware rewriter. It extracts multifaceted information from the chunk and formulates questions that encompass this information within a comprehensive context. This approach helps maintain semantic integrity and ensures that the generated atomic questions are meaningful and coherent. Below, for comparison, are the atomic questions generated from the same content.
> > >
> > >     1. "What are common methods or rules used to maintain semantic coherence when chunking text?"
> > >     2. "How do pronoun references get affected when text is divided into sub-chunks?"
> > >     3. "In what ways does the chronological order suffer when text is chunked?"
> > >     4. "What kind of causal relationships in text are most susceptible to disruption through chunking?"
> > >     5. "Are there any established techniques or tools that help in preserving semantic coherence in chunked texts?"
> > >     6. "What role do temporal adverbs play in maintaining the coherence of a text?"
> > >
> > > We observe that the generated atomic questions capture the relationships between chunk division and "disruption" from multiple perspectives, including "pronoun references," "chronological order," and "causal relationships." **Therefore, chunk atomization effectively extracts multifaceted information embedded within the chunk while preserving semantic coherence, thereby enhancing retrieval performance.**

---

> > > ### Author Response · Authors · 2024-11-28
> > > **Author Response to Reviewer vTwe (round 2, part2)**
> > >
> > > ### *Q2-3. Are there any experimental results for this point? For example, a variant of KAR³ atomizing chunks into plain texts instead of question?*
> > >
> > > Following the reviewer's suggestion, we conduct experiments using the sub-chunk setting on three datasets. This process involved separating the original chunks into sub-chunks with *spacy* library, defining each sentence of the original chunk as a sub-chunk, replacing the atomic questions with the newly parsed sentences, and then updating the selection prompt (listed in line 1350~1373 of the Appendix) to select the most useful sentence from the given list. The detailed experimental results are presented in Table 8, while Table 7 provides the count statistics of atomic question representation and plain-text representation. We can observe remarkable performance gaps between the atomization setting and the sub-chunking setting, which verifies the effectiveness of our proposed chunk atomizing.
> > >
> > > Table 8: Ablation study on atomic representation.
> > >
> > > | Variant                        | HotpotQA F1 | HotpotQA Acc | 2Wiki F1 |  2Wiki Acc | MuSiQue F1 | MuSiQue Acc |
> > > | ------------------------------ | ----------- | ------------ | -------- | ---------- | ---------- | ----------- |
> > > | KAR³ - plain text (Llamma 3)  |    62.61    |     73.60    |  52.90   |    58.72   |    45.88   |     54.20   |
> > > | KAR³ - plain text (GPT-4)     |    73.05    |     84.50    |   64.18  |    69.80   |    50.72   |     55.20   |
> > > | **KAR³ - question (Llama3)** |  **75.27**  |   **88.20** | **72.79** | **81.00** |  **50.68** |  **59.70** |
> > > | **KAR³ - question (GPT-4)**  |  **76.48**  |   **88.00** | **75.00** | **82.20** |  **57.86** |  **62.60** |
> > >
> > >
> > > ### *Q3.How do we establish the association between chunks with the atomic question as claimed in Line 283? It seems helpful to locate relevant atomic questions and link associated chunks.*
> > >
> > > To establish the association between chunks and atomic questions, we implement two vector stores: one for storing chunks and another for atomic questions. Each chunk is assigned a unique chunk ID, which serves as a foreign key linking the two stores. This approach allows us to maintain the richer information of the full chunk. During the retrieval phase, once we identify the relevant atomic questions, we can use their linked unique chunk IDs to retrieve the corresponding source chunks. This method effectively links the atomic questions to their associated chunks, as discussed in Line 283.
> > >
> > >
> > > ### *Q4. Is it the same LLM used for atomization and then prompted to select useful atomic questions? What’s the cost of the selector?*
> > >
> > > In the experimental results presented so far, the same LLM is used for the decomposer, selector, and generator components. The prompts designed for these components are detailed in Appendix A.6 (line 1296 ~ line 1392). It is worth noting that these components can be configured to use different language models, we leave it as future works (as discussed in line 535 ~ line 539 of our paper). The detailed token consumption of difference components on MuSiQue are illustrated in Table 9. The decomposition-selection loop iterates up to 5 rounds, leading to the multiple calls for decomposer and selector for each QA. Consequently, the decomposer and selector constitute the majority of the total consumption.
> > >
> > > Table 9: Token consumption (average/QA) of KAR³ on MuSiQue.
> > >
> > > | Component              | Prompt | Completion |  Total  |
> > > | ---------------------- | ------ | ---------- | ------- |
> > > | Decomposer             |  2691  |     768    |   3459  |
> > > | Selector               |  3278  |    1429    |   4707  |
> > > | Generator              |   556  |      98    |    654  |
> > > | KAR³                  |  6525  |    2295    |   8820  |
> > >
> > > In addition to the detailed token consumption for each component, Table 1 compares overall token consumption across various baselines. Our method uses fewer tokens than ProbTree and IRCoT and is comparable to Iter-RetGen. Importantly, our approach significantly outperforms these baselines in both F1 and Accuracy, highlighting its efficiency in balancing cost and performance.

---

> > > ### Author Response · Authors · 2024-12-01
> > > **Kind Reminder to Reviewer vTwe**
> > >
> > > Dear Reviewer vTwe,
> > >
> > > Thank you for reviewing our paper and sharing your thoughtful feedback. We have provided detailed responses to the additional questions raised in the comments, carefully addressing the key points you highlighted.
> > >
> > > As the rebuttal period approaches its conclusion, we sincerely welcome any further comments or concerns you may have. Your thoughtful feedback is deeply appreciated, and we hope our responses contribute to a more favorable assessment of our work.
> > >
> > > Thank you once again for your time and consideration.
> > >
> > > Best,
> > >
> > > Authors of submission 4468

---

> > > > ### Comment · Reviewer_vTwe · 2024-12-02
> > > >
> > > > I appreciate the detailed response from the authors.
> > > > Presently, there are so many new results in the response, which made me in a dilemma.
> > > > I would like to keep my current score.
> > > > Thank you again.

---

> > > > > ### Author Response · Authors · 2024-12-02
> > > > > **Clarifications on Reviewer Feedback and Additional Results**
> > > > >
> > > > > Dear Reviewer vTwe,
> > > > >
> > > > > Thank you for taking the time to review our responses and for your thoughtful feedback. We sincerely appreciate your acknowledgment of the detailed results we provided.
> > > > >
> > > > > We would like to emphasize that all the additional results included in our response were provided specifically to address the questions you raised. These results were not intended to introduce new content beyond your requests but rather to clarify and support the points in our original submission based on your valuable feedback.
> > > > >
> > > > > If there are any remaining concerns or aspects of the response that contribute to your dilemma, we would be more than glad to further address them. Your insights are invaluable to us, and we are committed to ensuring the submission fully meets your expectations.
> > > > >
> > > > > We hope these clarifications help, and we remain open to any additional questions or suggestions you may have. Thank you once again for your time and engagement with our work.
> > > > >
> > > > > Best regards,
> > > > >
> > > > > Authors of submission 4468

---

> ### Author Response · Authors · 2024-11-28
> **Author Response to Reviewer vTwe (round 2, part3)**
>
> ### *Q5. The Reason that IRCoT shows an improvement with LLaMA3 than with GPT4.*
>
> We appreciate the reviewer for highlighting the behavior of IRCoT. To investigate this, we analyzed the experimental logs of both models on the three datasets. IRCoT operates by iteratively prompting LLMs to generate rationale sentence based on query, previous rationale sentences and previous retrieved chunks. It then retrieves new passages with the newly generated rationale sentence. This process continues until the LLM can produce a final answer without needing additional rationale output.
>
> We count the number of reasoning rounds during the question-answering process for both the GPT and Llama versions, as shown in Table 10. Our analysis reveals that the IRCoT using GPT tends to resolve questions with fewer reasoning rounds, with 74.0%, 66.4%, 42.6% of questions answered without any reasoning for HotpotQA, 2Wiki, MuSiQue. Conversely, the IRCoT using Llama tends to engage in multiple rounds of reasoning, with 77.4%, 66.8%, 47.4% of questions requiring five reasoning rounds to produce answers for HotpotQA, 2Wiki, MuSiQue. **We suspect this is because GPT is more confident in its ability to provide direct answers, while Llama tends to offer reasoning steps rather than arriving at the answer in a single step.**
>
> Table 10: Reasoning round statistics of IRCoT on three datasets (500 questions per dataset).
>
> | Rounds | Hotpot: GPT      | Hotpot: Llama    | 2Wiki: GPT       | 2Wiki: Llama     | MuSiQue: GPT     |MuSiQue: Llama    |
> |--------|------------------|------------------|------------------|------------------|------------------|------------------|
> |   0    | 370 (74.0%)      |  11 (2.2%)       | 332 (66.4%)      |   4 (0.8%)       | 213 (42.6%)      |   3 (0.6%)       |
> |   1    |  76 (15.2%)      |   6 (1.2%)       |  83 (16.6%)      |  18 (3.6%)       |  71 (14.2%)      |  25 (5.0%)       |
> |   2    |  35 (7.0%)       |  34 (6.8%)       |  65 (13.0%)      |  55 (11.0%)      | 101 (20.2%)      |  80 (16.0%)      |
> |   3    |   4 (0.8%)       |  40 (8.0%)       |  10 (2.0%)       |  53 (10.6%)      |  36 (7.2%)       |  96 (19.2%)      |
> |   4    |   1 (0.2%)       |  22 (4.4%)       |   6 (1.2%)       |  36 (7.2%)       |  18 (3.6%)       |  59 (11.8%)      |
> |   5    |  14 (2.8%)       | 387 (77.4%)      |   4 (0.8%)       |  334 (66.8%)     |  61 (12.2%)      | 237 (47.4%)      |
>
> To provide a clearer understanding of the relationship between the number of reasoning rounds and performance changes, we calculated the average number of reasoning rounds for each question across both versions. The results are presented in Table 11. Additionally, the difference between the results of the two versions is indicated in parentheses following the Llama results, allowing the reviewer to easily observe the trend of changes.
>
> Table 11. Relations between the reasoning rounds and performance on three datasets
>
> | Rounds       | Hotpot: GPT      | Hotpot: Llama    | 2Wiki: GPT       | 2Wiki: Llama     | MuSiQue: GPT     |MuSiQue: Llama    |
> |--------------|------------------|------------------|------------------|------------------|------------------|------------------|
> |Avg. round/QA |     0.46         |   4.43 (+3.97)   |     0.57         |   4.42 (+3.85)   |       1.52       |   3.79 (+2.27)   |
> |   F1         |     67.3         |  74.59 (+7.29)   |    63.83         |  69.49 (+6.34)   |      47.57       |  43.12 (-4.45)   |
> |   Acc        |     81.00        |   88.0 (+7.00)   |    74.40         |  77.60 (+3.20)   |      49.20       |  49.20 (+0.40)   |
>
> As the difficulty levels of the three datasets differ significantly, with HotpotQA < 2Wiki < MuSiQue, the performance improvement on these datasets also varies. For simpler datasets like HotpotQA, Llama demonstrates a significant performance improvement compared to GPT. This is because Llama tends to activate reasoning processes to arrive at answers, while GPT is more prone to providing answers directly without reasoning.
> However, as the difficulty of the questions increases (e.g., in MuSiQue), GPT begins to activate its reasoning process rather than simply providing direct answers. This is reflected in the notable increase in the average reasoning rounds for GPT in MuSiQue (1.52) compared to HotpotQA (0.46). Conversely, Llama's reasoning rounds decrease (from 4.43 to 3.79) as question difficulty increases, which may indicate that Llama struggles with handling questions that require complex reasoning. When GPT activates its reasoning process, the performance advantage of Llama disappears, as evidenced by the results from MuSiQue.

---

### Official Review · Reviewer_1t8b · 2024-11-03

**Soundness:** 2
**Presentation:** 3
**Contribution:** 3
**Rating:** 5
**Confidence:** 3

**Summary:**

This paper introduces a novel Retrieval-Augmented Generation (RAG) framework called KAR-RAG, which enhances retrieval efficacy by incorporating a knowledge-aware task decomposition strategy. KAR-RAG decomposes complex queries into atomic questions, enabling more effective retrieval and logical reasoning from specialized datasets. The framework consists of four key components: a knowledge atomizer, query proposer, atomic retriever, and atomic selector, which collaboratively extract and utilize domain-specific knowledge to construct coherent rationales, achieving up to a 12.6% performance improvement over existing methods in multihop reasoning tasks.

**Strengths:**

1. This paper propose to enhance the RAG system with atomized knowledge retrieving mechnism. In particular, this paper designs knowledge atomizer, query proposer, atomic retriever, and atomic selector to construct the pipeline.
2. Experiments on complex question answering datasets show that the proposed method is effective.
3. The presentation of this paper is clear and easy to follow.

**Weaknesses:**

1. The effectiveness of each individual components should be further explored. For example, what if the knowledge is mingled together, rather than presented in atomic simple questions.
2. Lack of related works and baselines. This paper should survey and compare with more powerful RAG methods that elaborates on question decomposition, for example:
    - https://arxiv.org/pdf/2404.14464
    - https://arxiv.org/pdf/2406.19215

**Questions:**

See weakness

---

> ### Author Response · Authors · 2024-11-24
> **Author Response to Reviewer 1t8b**
>
> ### *Q1. The effectiveness of each individual component. What if the knowledge is mingled together, rather than presented in atomic simple questions.*
> In order to clarify the contribution of each component, we conduct ablation study in the main paper (line 483~521 and Table 2). We replace each proposed module with a simpler, standard module typically used in regular RAG systems. The results, presented in Table 5, demonstrate the effectiveness of each component. It is evident that each element is vital to our method's performance, particularly as the complexity of the questions increases, such as in the multi-hop scenarios seen with MuSiQue.
>
> Table 5. Ablation study on components of KAR³.
> | Variant Method               | Substitute Module   | HotpotQA F1 | HotpotQA Acc | 2Wiki F1 | 2Wiki Acc  | MuSiQue F1 | MuSiQue Acc |
> |------------------------------|---------------------|----------|------|-------|------|---------|------|
> | w/o Atomic Query Proposer    | Single Query        | 75.06    | 85.60| 70.19 | 76.40| 49.67   | 52.20|
> | **w/o Atomic Pair Retriever**    | **Chunk Retriever**     | 76.31    | 86.60| 67.14 | 72.40| 49.05   | 53.00|
> | w/o Atomic Selector          | Chunk Selector      | 72.80    | 83.20| 61.65 | 65.80| 49.31   | 53.40|
> | KAR³ **(Ours)**   | -                   | 76.48    | 88.00| 75.00 | 82.20| 57.86   | 62.60|
>
> If knowledge is mingled together without atomization, we can perform chunk-level retrieval to retrieve the relevant chunks for further reasoning. This approach is exactly how the variant method, "w/o Atomic Pair Retriever" is implemented, typically leading to suboptimal performance. For clarity and convenience, the corresponding row is highlighted in Table 5.
>
> ### *Q2. Lack of related work and baselines, such as SeaKR and ToR.*
>
> We appreciate the reviewer's suggestion to incorporate performance comparisons with two more baseline methods. In response, we have conducted a comprehensive evaluation against three new baselines, including ToR. The performance comparison results are presented in Table 2.
>
> It is important to note SeaKR relies on self-aware uncertainty calculated from hidden state embeddings of LLMs. The official implementation of SeaKR necessitates modifications to the inference function to access these embeddings, a requirement that is incompatible with GPT-4. Consequently, we have not included SeaKR in our comparison table.
>
> In the case of ToR, which has no available code, we compare our results directly with the performance reported in their paper. In addition to F1 metrics shown in Table 3, ToR reports supporting fact recall rates of 73.8, 79.4, and 48.5 for HotpotQA, 2Wiki, and MuSiQue, respectively. In contrast, our method achieves significantly higher supporting fact recall rates of 92.83, 94.06, and 73.8 on these datasets, as detailed in Table 6 of our paper.
>
> Overall, our method consistently outperforms these newly added baselines. For more comprehensive comparison results on both QA performance and token consumption, please refer to our responses to CQ1 and CQ2.

---

> ### Author Response · Authors · 2024-11-28
> **Authors' Kind Reminder to Reviewer 1t8b**
>
> Dear Reviewer 1t8b,
>
> Thank you for taking the time to review our paper and provide your valuable feedback. We have submitted detailed responses to your comments and suggestions, addressing the key points raised. **As the discussion is scheduled to conclude on December 2nd, we would greatly appreciate it if you could review our responses and offer any further clarifications at your earliest convenience. Your feedback is extremely important to us.**
>
> Thank you again for your time and effort.
>
> Best,
>
> Authors of Submission 4468

---

> > ### Author Response · Authors · 2024-12-01
> > **Kind Reminder to Reviewer 1t8b**
> >
> > Dear Reviewer 1t8b,
> >
> > Thank you for reviewing our paper and sharing your thoughtful feedback. We highly value any opportunities to address any potential remaining concerns before the discussion closes, which might be helpful for improving the rating of this submission. Please do not hesitate to comment upon any further concerns. Your feedback is extremely valuable to us.
> >
> > Thank you once again for your time and consideration.
> >
> > Best,
> >
> > Authors of submission 4468

---

### Official Review · Reviewer_4rFD · 2024-11-04

**Soundness:** 2
**Presentation:** 2
**Contribution:** 3
**Rating:** 6
**Confidence:** 4

**Summary:**

This paper introduces KAR-RAG, a novel RAG framework that improves retrieval efficacy through knowledge rewriting. Specifically, for each document chunk, KAR-RAG generates a set of atomic questions to label the available knowledge contained in the chunk, and use these atomic questions as indices for document chunk retrieval.

**Strengths:**

1. The paper is well-organized, and main components of the method (knowledge atomizer, query proposer, atomic retriever, atomic selector) are explained smoothly.
2. The paper proposes a novel rewrite-and-index approach to tackle retrieval-augmented generation.
3. Experiments conducted on both open-domain and domain-specific datasets demonstrate that KAR-RAG achieves state-of-the-art results.

**Weaknesses:**

1. The construction of KAR-RAG's atomic knowledge base can be highly cost-intensive, as generating atomic questions for each document chunk for a large text corpus leads to extensive LLM API consumption. The authors did not include a cost analysis to address this.
2. The paper’s motivation is a bit ambiguous. While the authors claim that the work focuses on enhancing “domain-specific knowledge mining” and “complex logical reasoning”, the paper lacks clear explanation on how KAR-RAG resolves these issues. Instead, it seems like KAR-RAG is a framework enhancing knowledge retrieval in general, showing no specific improvement over either “domain-specific knowledge mining” or “complex logical reasoning”.
3. The paper does not compare more recent and well-performing RAG baselines such as SearChain, ProbTree, etc, which may yield better results than KAR-RAG.

**Questions:**

When discussing the paper’s motivation, the authors could consider to address the challenge of accurate retrieval in general, instead of claiming to improve domain-specific knowledge retrieval (which may lead to misunderstandings on the paper’s core motivation).
Refered to Weaknesses session.

---

> ### Author Response · Authors · 2024-11-24
> **Author Response to Reviewer 4rFD**
>
> ### *Q1. Cost analysis on LLM API consumption.*
> To address reviewer's concern, we have conducted a comprehensive cost analysis to evaluate our model's API consumption compared to other baseline methods, and token consumption per QA of various methods are shown in Table 1. Our method consumes fewer tokens per QA compared to ProbTree and IRCoT, and is on par with Iter-RetGen. Notably, our approach significantly outperforms these baselines in both F1 and Accuracy metrics.
>
> Moreover, we implement our method and other baselines using the open-source LLM, meta-llama/Llama-3.1-70B-Instruct (including chunk atomization). The results are presented in Table 2, demonstrating that our method not only maintains excellent performance after replacing GPT-4 with an open-source LLM, but also significantly outperforms the other baselines. For a more detailed discussion, please refer to our response to CQ1.
>
> ### *Q2. How KAR-RAG resolves “domain-specific knowledge mining” and “complex logical reasoning”?*
> In our work, domain-specific knowledge pertains to information extracted from the corpus designated for RAG, which may not be fully covered by the pre-trained LLMs and can sometimes diverge from LLMs' intrinsic knowledge. For instance, within a particular domain, "ice cream" might be used as a unique project name rather than referencing the typical frozen dessert. This domain-specific knowledge serves as external knowledge to assist LLMs in generating responses that are consistent with the knowledge from the RAG corpus. **Nevertheless, integrating domain-specific knowledge into the logical reasoning process poses challenges, because it require RAG system to a) effectively extract domain-specific knowledge, b) precisely retrieve it relevant knowledge via queries, and c) successfully construct coherent reasoning chain based on it. To address these challenges, we propose KAR³-RAG encompassing knowledge atomization, atomic pair retrieval, and query decomposition with reference context.** Specially, our method employ the reference context as known domain-specific knowledge to influence the subsequent query decomposition, thereby tailoring the reasoning path according to the available corpus.
> In order to assist reviewer understand our approach, we present an example in Table 4, and detailed comparison between w/ and w/o domain-specific knowledge is provided in the response to CQ3.
>
> In the revised version, we will clarify the concept of domain-specific knowledge and delineate the challenges encountered when engaging in domain-specific logical reasoning to more clearly convey our motivation and highlight our contribution.
>
> ### *Q3. Lack of related work and baselines, such as SearChain, ProbTree .*
> We appreciate the reviewer's suggestion to incorporate performance comparisons with two more baseline methods. We have conducted a comprehensive evaluation against three new baselines, including SearChain, ProbTree. The performance comparison results are presented in Table 3. Furthermore, we have analyzed the token consumption for these baselines and present the results in Table 1. Our method demonstrates superior performance over these recently evaluated baselines. For a more exhaustive set of comparison results encompassing a range of baselines, please refer to the detailed responses to Comments CQ1 and CQ2.

---

> ### Author Response · Authors · 2024-11-28
> **Authors' Kind Reminder to Reviewer 4rFD**
>
> Dear Reviewer 4rFD,
>
> Thank you for taking the time to review our paper and provide your valuable feedback. We have submitted detailed responses to your comments and suggestions, addressing the key points raised. **As the discussion is scheduled to conclude on December 2nd, we would greatly appreciate it if you could review our responses and offer any further clarifications at your earliest convenience. Your feedback is extremely important to us.**
>
> Thank you again for your time and effort.
>
> Best,
>
> Authors of Submission 4468

---

> ### Author Response · Authors · 2024-12-01
> **Kind Reminder to Reviewer 4rFD**
>
> Dear Reviewer 4rFD,
>
> Thank you for reviewing our paper and sharing your thoughtful feedback. We highly value any opportunities to address any potential remaining concerns before the discussion closes, which might be helpful for improving the rating of this submission. Please do not hesitate to comment upon any further concerns. Your feedback is extremely valuable to us.
>
> Thank you once again for your time and consideration.
>
> Best,
>
> Authors of submission 4468

---

### Official Review · Reviewer_qWux · 2024-11-04

**Soundness:** 3
**Presentation:** 3
**Contribution:** 3
**Rating:** 6
**Confidence:** 3

**Summary:**

This paper describes an approach to enhance and use knowledge in RAG setting to improve the overall of reasoning performance of LLMs. Paper proposes atomiser, a question generation for a given query/chunk. It uses the questions generated from the query to generate questions using LLM and use it in retriever to retrieve the context chunks. These are further used to generate any missing context by iteratively decomposing queries to gather mode suitable context to answer the given query and the final answer is generated using this process.

**Strengths:**

Paper is easy to follow and authors have done good job in explaining the overall method with good examples. Method shows consistent improvements across all the chosen benchmarks which require reasoning. Another strong point I see is knowledge augmentation is happening with query decomposition and context gathering, rather than going through KG generation etc, which can be more involved and can introduce another place for injecting errors.

**Weaknesses:**

Authors have no talked about the overhead this method introduces in terms of generating the final answer in terms of response time/ cost of using LLM in the whole process.

**Questions:**

Do any of the existing query decomposition techniques be used as is without using LLM here ? For example for the of interest in example presented in Figure 2, main questions needed are who is the direct of the film and who is the mother or parent of person X . Can't this be done using other methods ?

---

> ### Author Response · Authors · 2024-11-24
> **Author Response to Reviewer qWux**
>
> ### *Q1. Overhead this method introduces in terms of generating the final answer in terms of response time/cost of using LLM in the whole process.*
>
> The LLM API usage in our method consists of two main components: 1) chunk atomization as a one-time preprocessing step, and 2) the inference process, which includes query decomposition, retrieval, selection, and generation.
>
> For preprocessing, the API consumption scales linearly with the number of data chunks and constitutes an overhead that varies across different benchmarks. Specifically, the average token usage per chunk for the three benchmarks is 338 tokens for HotPotQA, 321 tokens for 2Wiki, and 320 tokens for MuSiQue.
>
> Regarding inference costs, we conduct a thorough cost analysis to compare our model's API usage with other baseline methods. As shown in Table 1, our method consumes fewer tokens per QA compared to ProbTree and IRCoT, and is on par with Iter-RetGen. Notably, our approach outperforms these baselines in both F1 and Accuracy metrics by a considerable margin. Given the variability in GPT-4 API response times, we use token consumption per QA as a measure of cost.
>
> To further address the reviewer's concern on the API cost, we implement our method and other baselines using the open-source LLM, meta-llama/Llama-3.1-70B-Instruct (including chunk atomization). The results presented in Table 2, demonstrate that our method not only sustains excellent performance after replacing GPT-4 with an open-source LLM, but also significantly outperforms the other baselines. For a more detailed discussion, please refer to our response to CQ1.
>
>
> ### *Q2. Do any of the existing query decomposition techniques without using LLM?*
>
> There are alternative methods for decomposing questions into sub-questions without relying on LLMs. Earlier approaches used a combination of hand-crafted heuristics, rule-based algorithms, and supervised learning for decompositions. For instance, Ethan introduced the One-to-N Unsupervised Sequence Transduction(ONUS) approach, which employs Seq2Seq models to map complex, multi-hop questions to simpler, single-hop sub-questions.
>
> In comparison with earlier methods for decomposing questions, LLMs offer significant advantages. LLMs capture a wide range of linguistic patterns and knowledge, enabling them to generalize better to unseen questions and adapt to various contexts. They leverage pre-trained knowledge to provide more accurate and relevant decompositions with less manual intervention. Although approaches such as Self-Ask and SeaChain similarly employ LLMs to sequentially generate sub-questions, they fall short in adequately harnessing the knowledge contained within RAG data. This leads to a deficiency in generating sub-questions that are contextually pertinent to the RAG data.

---

> > ### Author Response · Authors · 2024-12-01
> > **Kind Reminder to Reviewer qWux**
> >
> > Dear Reviewer qWux,
> >
> > Thank you for reviewing our paper and sharing your thoughtful feedback. We highly value any opportunities to address any potential remaining concerns before the discussion closes, which might be helpful for improving the rating of this submission. Please do not hesitate to comment upon any further concerns. Your feedback is extremely valuable to us.
> >
> > Thank you once again for your time and consideration.
> >
> > Best,
> >
> > Authors of submission 4468

---

> ### Author Response · Authors · 2024-11-28
> **Authors' Kind Reminder to Reviewer qWux**
>
> Dear Reviewer qWux,
>
> Thank you for taking the time to review our paper and provide your valuable feedback. We have submitted detailed responses to your comments and suggestions, addressing the key points raised. **As the discussion is scheduled to conclude on December 2nd, we would greatly appreciate it if you could review our responses and offer any further clarifications at your earliest convenience. Your feedback is extremely important to us.**
>
> Thank you again for your time and effort.
>
> Best,
>
> Authors of Submission 4468

---

### Author Response · Authors · 2024-11-24
**General Responses and Common Questions (part 3)**

### CQ3. Lack of specific explanation over domain-specific knowledge and complex logical reasoning upon them. (for 4rFD and vTwe)

In our work, domain-specific knowledge pertains to information extracted from the corpus designated for RAG, which may not be fully covered by the pre-trained LLMs and can sometimes diverge from LLMs' intrinsic knowledge. For instance, within a particular domain, "ice cream" might be used as a unique project name rather than referencing the typical frozen dessert. This domain-specific knowledge serves as external knowledge to assist LLMs in generating responses that are consistent with the knowledge from the RAG corpus. **Nevertheless, integrating domain-specific knowledge to domain-aligned reasoning and generation introduces significant challenges, which include the difficulties of a) effectively extracting domain-specific knowledge, b) precisely retrieving relevant domain-specific knowledge with queries, and c) successfully constructing coherent reasoning chain based on it. To tackle these challenges, we propose a novel approach encompassing knowledge atomization, atomic pair retrieval, and query decomposition with reference context.**

Specially, our method employ the reference context as known domain-specific knowledge to influence the subsequent query decomposition, thereby tailoring the reasoning path according to the available corpus. To illustrate this approach, Table 4 presents an example to aid reviewers in understanding this concept. In this example, *A* and *B* are individuals with common names, which may confuse LLMs and may lead them to generate the information from the other person with the same name. *C* represents a company name.

Table 4: Query Example for query decomposition and logical reasoning .

| **Query**         | "Who joined the company *C* first, *A* or *B*?" |
| ----------------- | ---------------- |
| **Supporting facts** | *S1*. "After graduating from University, *A* found his first job as a product manager in *C*."  |
|      | *S2*. "As the company underwent spin-off litigation, *A* experienced uncertainty about the company's future and carefully considered his career options. |
|      | *S3*. "After a period of corporate turmoil caused by the spin-off lawsuit, *B* was invited to join *C* as a strategic consultant to help navigate the challenges and capitalize on new opportunities." |

Referring to the **part one** of example analysis below, given a query, LLMs that rely on common knowledge tend to decompose it into two straightforward subqueries often overlooking the availability of corresponding answers in the RAG data. Although two related facts *S1* and *S3* are retrieved, the provided information is insufficient to construct a reasoning chain needed to answer the question.
Our method, illustrated in the **part two** of example analysis, incorporates the reference context from previously retrieved data to guide the subsequent query decomposition. This approach generates potential reasoning paths based on the domain-specific knowledge retrieved from RAG data, and further utilize subqueries to retrieve the relevant knowledge, ultimately arriving at the correct answer with a complete reasoning chain.

In the revised version, we will elucidate the definition of domain-specific knowledge and delineate the challenges encountered when engaging in domain-specific logical reasoning to more clearly convey our motivation and highlight our contribution.

**Example analysis**

**Part one: Decompose and Reasoning with *common knowledge***

*Decompose query without context*:

    1. "When does A join the company C?"
    2. "When does B join the company C?"

*Retrieve*: *S1* and *S3*

*Reasoning*: insufficient information to answer.


**Part two: Decompose and Reasoning with *domain-specific knowledge***

*Context retrieved from previous iterations*: *S1* and *S3*

*Decompose query with context*:

    1. "Did A join the company while C was undergoing spin-off proceedings?"
    2. "When did A graduate from university?"
    3. "When did C become involved in a spin-off litigation?"

*Retrieve*: *S2*

*Reasoning*: Based on *S1*, *S2*, and *S3*, *A* joined *C* before the spin-off lawsuit, while *B* joined *C* after the spin-off lawsuit. Therefore, the answer is *A*.

---

### Author Response · Authors · 2024-11-24
**General Responses and Common Questions (part 2)**

### CQ2. Comparison with more baselines. (for reviewer 4rFD and 1t8b)

We appreciate the reviewers' suggestion to incorporate performance comparisons with four additional baseline methods. In response, we have conducted a comprehensive evaluation against three of these baselines: SearChain, ProbTree, and Tree of Reviews (ToR). As shown in Table 3, our method consistently outperforms these baselines.

It is important to note that the baseline, SeaKR, relies on self-aware uncertainty calculated from hidden state embeddings of LLMs. The official implementation of SeaKR necessitates modifications to the inference function to access these embeddings, a requirement that is incompatible with GPT-4. Consequently, we have not included SeaKR in our comparison table. For the other works with available code (SearChain and ProbTree), we conduct experiments using our sampled datasets while making every effort to adhere to their default settings. And the metrics reported in Table 3 were calculated using our evaluator, which normalizes answers before evaluation. In the case of ToR, which has no available code, we compare our results directly with the performance reported in their paper. In addition to F1 metrics shown in Table 3, ToR reports supporting fact recall rates of 73.8, 79.4, and 48.5 for HotpotQA, 2Wiki, and MuSiQue, respectively. In contrast, our method achieves significantly higher supporting fact recall rates of 92.83, 94.06, and 73.8 on these datasets, as detailed in Table 6 of our paper.

Table 3. Performance comparison on multi-hop QA datasets. Best in bold.

| Method                | HotpotQA F1     | HotpotQA Acc    | 2Wiki F1        | 2Wiki Acc       | MuSiQue F1      | MuSiQue Acc     |
| --------------------- | --------------- | --------------- | --------------- | --------------- | --------------- | --------------- |
| Zero-Shot CoT         | 43.94           | 53.60           | 41.40           | 43.87           | 22.90           | 23.47           |
| Naive RAG             | 72.67           | 82.60           | 59.74           | 62.80           | 43.31           | 44.40           |
| Self-Ask w/ Retrieval | 71.40           | 80.00           | _69.06_       | _75.00_       | 46.76           | 51.40           |
| IRCoT                 | 67.30           | 81.00           | 63.83           | 70.40           | 47.57           | 49.20           |
| Iter-RetGen           | _75.27_       | _86.60_       | 67.21           | 73.60           | _52.48_       | _55.60_       |
| SeaChain\[1\]         | 40.48           | 74.40           | 15.67           | 68.40           | 33.26           | 45.80           |
| ProbTree\[2\]         | 62.41           | 73.40           | 69.42           | 80.00           | 47.01           | 54.20           |
| ToR\[3\]              | *63.1*        | *-*           | *62.9*        | *-*           | *43.6*        | *-*           |
| **KAR³ (Ours)**  | **76.48** | **88.00** | **75.00** | **82.20** | **57.86** | **62.60** |

[1]. **SeaChain**: Since the official code loads pre-trained models from local without uploading those models online, we find models with most-similar name from HuggingFace to adapt it. Besides, we use *BAAI/bge-m3* instead of the ColBERT retriever due to environmental issues.

[2]. **ProbTree**: The official code was executed with bug fixes.

[3]. **ToR**: No official code has been released for ToR. The performance metrics provided are those reported by the authors using GPT-4-Turbo. Please note that the sampled testing suite and the method for answer normalization before F1 calculation may differ from ours.

---

### Author Response · Authors · 2024-11-24
**General Responses and Common Questions (part 1)**

We appreciate the valuable feedback and insightful comments from all reviewers. In this rebuttal, we will address all reviewer comments point by point and revise our paper accordingly.

### CQ1. Cost analysis on LLM API Consumption and implementation with open-source LLMs (for reviewer qWux, 4rFD and vTwe)

We have conducted a comprehensive cost analysis to evaluate our model's API consumption compared to other baseline methods, and token consumption per QA of various methods are shown in Table 1. Our analysis primarily focuses on the inference cost, as chunk atomization is a one-time preprocessing step. The LLM API consumption for this preprocessing scales linearly with the number of data chunks and constitutes an overhead that varies across different benchmarks. Specifically, the average token usage per chunk for the three benchmarks is 338 tokens/chunk (HotPotQA), 321 tokens/chunk (2Wiki), and 320 tokens/chunk (MuSiQue).

As demonstrated in Table 1, our method utilizes fewer tokens than both ProbTree and IRCoT, and is comparable to Iter-RetGen. However, our approach significantly outperforms these baselines on both F1 and Accuracy by a considerable margin. This demonstrates the efficiency of our approach in balancing cost and performance. It is important to highlight that our method focuses on exploring potential reasoning chains, necessitating a thoughtful analysis generation during query decomposition with context at each iteration. As a result, completion token usage constitutes approximately one-quarter of the total consumption, distinguishing our approach from other baselines.

Table1: Token consumption (average/QA) and performance comparison on MuSiQue.

| Method                 | Prompt | Completion | Total ⬇️  | F1 ⬆️    | Acc ⬆️   |
| ---------------------- | ------ | ---------- | --------- | ----------| ----------|
| Zero-Shot CoT          | 85     | 105        | 191       | 22.90     | 23.47     |
| Naive RAG              | 1765   | 103        | 1869      | 43.31     | 44.40     |
| Self-Ask w/ Retrieval  | 5894   | 619        | 6514      | 46.76     | 51.40     |
| IRCoT                  | 9703   | 86         | _9789_    | 47.57     | 49.20     |
| Iter-RetGen            | 8140   | 473        | 8614      | _52.48_   | _55.60_   |
| SeaChain               | 2247   | 627        | 2875      | 33.26     | 45.80     |
| ProbTree               | 25225  | 650        | **25875** | 47.01     | 54.20     |
| **KAR³ (Ours)**        | 6525   | 2295       | 8820      | **57.86** | **62.60** |

To further address reviewers' concerns regarding API costs, we implement our method and other baselines using the open-source LLM, meta-llama/Llama-3.1-70B-Instruct (including chunk atomization). The results, presented in Table 2, indicate that this implementation not only maintains excellent performance after replacing GPT-4 with an open-source LLM, but also significantly outperforms the other baselines.

Table 2: Performance comparison with different LLMs.

| Method | HotpotQA F1 | HotpotQA Acc | 2Wiki F1 | 2Wiki Acc | MuSiQue F1 | MuSiQue Acc |
| ----------------------------- | ----- | ----- | ----- | ----- | ----- | ----- |
| Zero-Shot CoT(Llama3)         | 40.10 | 54.80 | 38.54 | 43.20 | 15.69 | 19.80 |
| Naive RAG(Llama3)             | 70.78 | 84.20 | 56.58 | 62.20 | 32.53 | 36.40 |
| Self-Ask w/ Retrieval(Llama3) | 70.25 | 83.00 | 66.25 | 74.00 | 38.19 | 44.20 |
| IRCoT(Llama3)                 | 74.59 | 88.00 | 69.49 | 77.60 | 43.12 | 49.60 |
| Iter-RetGen(Llama3)           | 72.23 | 85.20 | 59.21 | 65.00 | 37.16 | 40.40 |
| **KAR³(Llama3)**       | **75.27** | **88.20** | **72.79** | **81.00** | **50.68** | **59.70** |
| **KAR³(GPT-4)**        | **76.48** | **88.00** | **75.00** | **82.20** | **57.86** | **62.60** |

We will incorporate the results from Tables 1 and 2 into the revised version to provide a more comprehensive view of the token consumption of our work and its performance with open-source LLMs.

---

### Meta-Review · Area_Chair_29X5 · 2024-12-22

**Metareview:**

The paper introduces KAR, a novel RAG framework combining knowledge atomization, query decomposition, and retrieval optimization to enhance domain-specific reasoning in multi-hop QA. Experiments on HotpotQA, 2Wiki, and MuSiQue benchmarks show improvements over baselines.

Strengths:
-The framework is well-designed, with innovative components like the knowledge atomizer and atomic selector, as noted by reviewers (qWux, 4rFD).
-The paper is well-organized and well written
-The method shows clear empirical advantages.

Weaknesses:
-Cost-Intensive construction: Several reviewers (4rFD, vTwe, and qWux) highlight concerns about the high computational costs of constructing KAR's atomic knowledge base, which relies heavily on repeated LLM calls.
-Ambiguous Motivation: Reviewers (4rFD and vTwe) note that the motivation for the paper is unclear, particularly in explaining how the proposed method effectively addresses the stated challenges of domain-specific reasoning and retrieval.
-Missing Baseline Comparisons: Reviewers (4rFD and 1t8b) emphasize the need for additional comparisons with strong RAG baselines.
-Component Effectiveness: The reviewers raised concerns about lack of ablations on different components of the method (1t8B).

Despite strong empirical results, there are several concerns regarding the experimental rigor and presentation of the work.

**Additional Comments On Reviewer Discussion:**

The rebuttal provided a thorough analysis of cost considerations and showed effectiveness with more affordable LLMs, and resolved ambiguities about motivation. The rebuttal also presented additional experimental results to address identified gaps. Although these additions strengthened the work, reviewers maintained reservations about scalability issues, depth of technical contributions, and methodological straightforwardness. Moreover, while the substantial number of new experiments in the rebuttal enhanced the submission, it also emphasized the paper's initial incompleteness and suggested the need for significant revision.

---

### Decision · Program_Chairs · 2025-01-22

Reject